

# Unsupervised classification identifies coherent thermohaline structures in the Weddell Gyre region

Dan(i) Jones[1], Maike Sonnewald[2,3,4], Shenjie Zhou[1], Ute Hausmann[1], Andrew J.S. Meijers[1], Isabella Rosso[5,6], Lars Boehme[7], Michael P. Meredith[1], and Alberto C. Naveira Garabato[8]

[1]British Antarctic Survey, NERC, UKRI, Cambridge, UK
[2]Princeton University, Princeton, NJ, USA
[3]NOAA Geophysical Fluid Dynamics Laboratory, Princeton, NJ, USA
[4]University of Washington, Seattle, WA, USA
[5]Scripps Institution of Oceanography, UCSD, La Jolla, CA, USA
[6]GeoOptics Switzerland SA, Lausanne, Switzerland
[7]SMRU, University of St. Andrews, UK
[8]University of Southampton, Southampton, UK

**Correspondence:** D.C. Jones (dannes@bas.ac.uk)

**Abstract.** The Weddell Gyre is a major feature of the Southern Ocean and an important component of the planetary climate system; it regulates air-sea exchanges, controls the formation of deep and bottom waters, and hosts upwelling of relatively warm subsurface waters. It is characterized by extremely low sea surface temperatures, ubiquitous sea ice formation, and widespread salt stratification that stabilises the water column. Observing the Weddell Gyre is challenging, as it is extremely remote and largely covered with sea ice. At present, it is one of the most poorly-sampled regions of the global ocean, highlighting the need to extract as much value as possible from existing observations. Here, we apply a profile classification model (PCM), which is an unsupervised classification technique, to a Weddell Gyre profile dataset to identify coherent regimes in temperature and salinity. We find that, despite not being given any positional information, the PCM identifies four spatially coherent thermohaline domains that can be described as follows: (1) a circumpolar class, (2) a transition region between the circumpolar waters and the Weddell Gyre, (3) a gyre edge class with northern and southern branches, and (4) a gyre core class. PCM highlights, in an objective and interpretable way, both expected and under-appreciated structures in the Weddell Gyre dataset. For instance, PCM identifies the inflow of Circumpolar Deep Water (CDW) across the eastern boundary, the presence of the Weddell-Scotia Confluence waters, and structured spatial variability in mixing between Winter Water and CDW. PCM offers a useful complement to existing expertise-driven approaches for characterising the physical configuration and variability of the Weddell Gyre and surrounding regions.

## 1 Introduction

The Southern Ocean is a key region in the global climate system, hosting crucial transformations that supply waters to both the upper and lower limbs of the global ocean overturning circulation (IPCC, 2022, Ch. 3). The lower limb is renewed by dense waters that form and are exported northward, flooding the majority of the global abyssal ocean (Johnson, 2008). The





Weddell Sea is important for this dense water production and export, with its southern and western continental shelves hosting interactions with floating ice shelves, as well as strong cooling and sea ice production in polynyas (Vernet et al., 2019). These exchanges result in shelf waters that are extremely cold (some below the surface freezing point) and comparatively saline; this gives them sufficient density to spill from the shelf down the slope and into the deep Weddell Sea, entraining mid-depth waters as they descend (Foster and Carmack, 1976; Gill, 1973; Killworth, 1983; Gordon et al., 2001).

In addition to this mode of deep-ocean ventilation, sporadic occurrences of deep convection over the deep Weddell Sea have been observed, especially in the vicinity of Maud Rise. Here, large-scale polynyas can emerge that enable dense water production and sinking; this was first noted in the 1970s (Gordon, 1978), with indications that this may have recently recurred after a decades-long hiatus (Campbell et al., 2019). The dense waters that form in the Weddell Sea penetrate northwards to supply the lower limb of the Atlantic Meridional Overturning Circulation. There are signs that this export is dwindling in

recent years (Johnson et al., 2008), though hiatuses in the decline have been noted (Abrahamsen et al., 2019). To reach the Atlantic, the dense water must navigate the complex bathymetry of the Scotia Arc, the southern flank of which comprises the South Scotia Ridge. The most direct route for dense water to cross this ridge is Orkney Passage (Naveira Garabato et al., 2002), though the possibility of significant outflow around the outside of the Scotia Arc also exists (Jullion et al., 2014).

## 1.1   Circulation and water masses in the Weddell Gyre region

The Weddell Gyre region is a complex nexus of circumpolar and gyre circulation, with ubiquitous water mass formation, transport, and destruction (Fig. 1). Horizontal circulation to the north of the gyre is dominated by the Antarctic Circumpolar Current (ACC), the eastward-flowing current system that comprises several discrete fronts and which is characterised by strong spatial variability (Sokolov and Rintoul, 2009; Rintoul and Garabato, 2013). The Weddell Gyre separates the ACC from Antarctica in the Atlantic sector; it is a cyclonic circulation system that extends east from the eastern Antarctic Peninsula. No

topographic or distinct current feature forms its eastern extent, but it is nominally set to approximately 30°E, 70°E, or further eastwards. In the meridional direction, it extends from the continental slope to approximately 60° (Fahrbach et al., 1994; Park et al., 2001; Meijers et al., 2010; Vernet et al., 2019). At its eastern flank, the voluminous mid-depth Circumpolar Deep Water (CDW) from the ACC is entrained into the Weddell Gyre, where it mixes to become cooler and fresher, and is usually termed Weddell Deep Water or Warm Deep Water (WDW) (Fahrbach et al., 1994). This is the oceanic source that penetrates onto the

shelf and is modified to become the dense waters that ultimately flow northward at depth (Jullion et al., 2014; Naveira Garabato et al., 2016). Circulation within the gyre exhibits a two-cell structure, with the western cell centred around 40°W and the eastern gyre centred around 18°E (Reeve et al., 2019). The gyre is forced by westerly winds over its northern edges, producing upwelling in its center; it is also forced by easterly winds over its southern edges, producing downwelling along its southern limb (Naveira Garabato et al., 2016). In addition, it is subject to strong buoyancy forcing. Separating the ACC to the north from

the Weddell Gyre to the south, the Weddell-Scotia Confluence (WSC) follows close to the complex bathymetry of the South Scotia Ridge (Fig. 1). The WSC is identifiable from the waters on either side by characteristically low stratification at mid depths; this has been ascribed to the injection and sinking of shelf waters from the tip of the Antarctic Peninsula (Whitworth et al., 1994, Fig. 1).



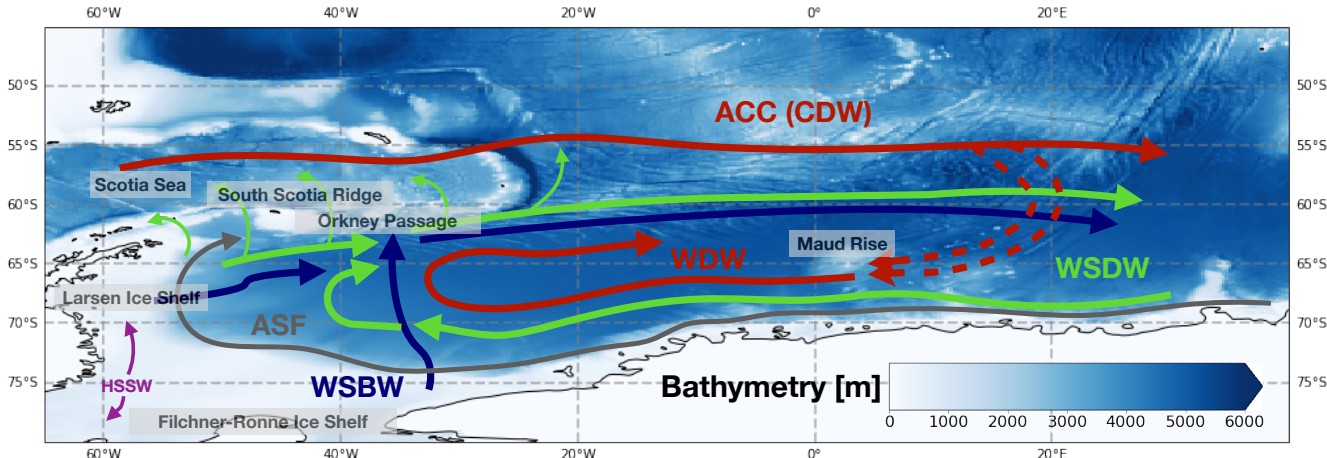

**Figure 1.** Schematic of the circulation of the Weddell Gyre region. The broad circulation features include the Antarctic Circumpolar Current (ACC), Circumpolar Deep Water (CDW), Weddell Sea Deep Water (WSDW), Warm Deep Water (WDW), Weddell Sea Bottom Water (WSBW), High Salinity Shelf Water (HSSW), and the Antarctic Slope Front (ASF). Selected geographic features are named, and the colour scale shows the bathymetry. Adapted from Vernet et al. (2019).

Despite the importance of the Weddell Gyre and its surroundings, its structure and dynamics are not thoroughly understood, partly because of the practical difficulties in observing the region. It is remote and inaccessible, especially in winter, when it is extremely challenging to reach by ship, and it is frequently covered by clouds, making it challenging to acquire complete satellite observations in the visible spectrum. These limitations underscore the importance of making the most of the sparse data that we do have, using a wide variety of observational analysis and reanalysis techniques (e.g. Reeve et al. (2016)).

### 1.2 Unsupervised classification of oceanographic variables

In recent years, researchers have employed unsupervised classification, a broad suite of methods for finding coherent patterns in unlabelled data, to a variety of oceanographic problems. For example, Sonnewald et al. (2019) identified coherent dynamical regimes in the global ocean by using the terms of the barotropic vorticity budget as the "features" or "dimensions" of the unsupervised classification analysis. Jones and Ito (2019) applied a similar method to the surface carbon budget in a numerical model, and Sonnewald et al. (2020) used a multi-layered unsupervised classification approach to identify ecological regimes. Couchman et al. (2021) applied unsupervised learning to cluster fluid patches according to their background buoyancy frequency and turbulent dissipation rates. Because the "decisions" made by many of these unsupervised classification approaches are relatively transparent and can be thoroughly analysed, at least when the complexity of the model is kept at a manageable level, unsupervised classification results are often interpretable by oceanographic experts, highlighting their potential for uncovering novel or underappreciated structures in complex, multi-dimensional oceanographic data (Sonnewald and Lguensat, 2021). Generally speaking, unsupervised classification can be thought of as a "hypothesis generation tool" (Kaiser et al., 2022).



For a review of recent machine learning advances in oceanography, including unsupervised classification, see Sonnewald et al. (2021).

In many oceanographic unsupervised classification applications, the dataset consists of a collection of profiles, where a "profile" refers to a set of measurements taken at various pressures at a single location in latitude and longitude. The measured quantities may consist of temperature, salinity, and biogeochemical variables such as oxygen. Often the applications revolve around identifying different "profile types", which may also be called classes or groups. There are a variety of unsupervised classification approaches for working with profile data. For example, Thomas and Müller (2022) used a self-organising map approach to group temperature profiles in the European Arctic, in part to facilitate numerical model validation. One particular method is the profile classification model (PCM) approach, which is an ocean-specific application of Gaussian mixture modelling (GMM), for identifying the profile types (Maze et al., 2017). In PCM, one attempts to statistically model the profiles, typically as represented in an abstract principal component space, as a collection of multi-dimensional Gaussian functions. The result is a set of profile types that feature similar vertical structures across one or more measured quantities (e.g. with temperature and salinity data both contributing to the identification of the profile types).

PCM has been used in a number of applications in recent years. Jones et al. (2019) applied the PCM approach to Southern Ocean temperature profile data, identifying spatially coherent regimes that roughly align with modern understanding of Southern Ocean fronts and subtropical structure. PCM was able to identify these regions despite the fact that it was not given any location information about the profiles. Rosso et al. (2020) expanded this analysis to include salinity in the Indian sector of the Southern Ocean, identifying frontal zones and the variability of water masses present in each zone. Houghton and Wilson (2020) applied PCM to Pacific Ocean temperature data, connecting the temporal evolution of the tropical classes to ENSO, thereby deriving a novel ENSO proxy. Sambe and Suga (2022) applied PCM to the Northwest Pacific Ocean, allowing them to link the variability of the Kuroshio extension to the regional vertical structure. Recently, Xia et al. (2022) used PCM to identify three different types of Antarctic Intermediate Water (AAIW) and their formation regions. PCM may also be useful for finding circulation pathways; Boehme and Rosso (2021) used PCM to identify separate warm and cold modes of transport on the Amundsen Sea shelf. Desbruyères et al. (2021) used the approach, called ocean profile clustering in their paper, to separate subpolar waters from subtropical waters in the North Atlantic, enabling a data-driven analysis of the cooling-to-warming transition of the Subpolar North Atlantic. PCM has also been applied to temperature profile quality control (Zhang et al., 2022).

Chapman et al. (2020) challenged oceanographic researchers to rethink our treatment of boundaries and fronts, arguing that a more locally-tailored, application-specific, and perhaps even probabilistic approach may be more suitable than attempting to identify a single global set of boundaries between oceanic thermohaline and biogeochemical structures. Because PCM identifies profile types in a probabilistic way, returning a set of probabilities across the profile types, it can be used to reframe how we describe boundaries between oceanographic structures. To this end, Thomas et al. (2021) introduced a method for defining boundaries in a probabilistic fashion using PCM. This quantity, called the inter-class comparison metric (I-metric), uses the difference between the class with the highest probability and the runner-up to quantify the likelihood that a given profile is on the boundary between two classes. Given the ambiguity of the precise extent of the Weddell Gyre, the I-metric approach is likely to be a suitable framework for rethinking the boundaries between components of the gyre and its surroundings.





The structure of this papers is as follows. In section 2, we describe the profile dataset and the unsupervised classification method used to identify coherent structures in the data. In section 3, we examine the distribution properties across the four-class PCM, including how those properties vary seasonally. Finally, in section 4, we discuss these results in the context of the literature on the Weddell Gyre and its environs.

## 2 Data and methods

In this section, we describe the profile dataset analysed in our study (section 2.1) and the unsupervised classification approach used to identify coherent thermohaline structures within the profile dataset (section 2.2). For details, see appendix B.

### 2.1 Profile dataset

For this study, we used a set of temperature and salinity profiles from the Atlantic and Indian sectors of the Southern Ocean. The dataset consists of profiles taken by Argo floats and ship-based CTDs as recorded in the World Ocean Database within a box defined by 85°S – 30°S, 65°W – 80°E. We only consider profiles with good position and time flags, as well as good temperature, salinity, and pressure measurements with good flags. Duplicated profiles are identified when multiple profiles are found within 24 hours over the same 2 km×2 km grid cell, following Schmidtko et al. (2014), and only one profile within the spatio-temporal window is used. We then used the MITprof toolbox to pre-process the selected profiles, re-gridding them onto standard pressure levels (Forget et al., 2015; Forget, 2017). This step is necessary because one requirement of PCM/GMM is that the data have a consistent number of "features", also referred to as "dimensions" in machine learning terminology, throughout the entire dataset. Here, we select a standard pressure grid with 72 vertical levels, with fine enough vertical resolution to preserve temperature and salinity structure while avoiding data gaps at depth at each grid point. The vertical interval varies from 20 dbar at the surface to 100 dbar in the deep ocean. In total, we used 188,885 Argo profiles and 34,915 ship-based CTD profiles covering 20-1000 dbar for the initial classification step, i.e. the identification of near-Antarctic waters.

We first used PCM to sort the entire profile dataset into five classes based on their combined temperature and salinity structure. Broadly speaking, these classes may be described as (1) a subtropical Atlantic sector class, (2) a subtropical Indian sector class, (3) a more northern circumpolar class, (4) a more southern circumpolar class, and (5) a near-Antarctic class that roughly sits south of the Polar Front (Appendix A). One could obtain a similar sub-Antarctic class by simply selecting all profiles located south of the Polar Front. Because we are primarily interested in the Weddell Gyre and its surroundings, we chose to focus on the near-Antarctic class. Effectively, the rest of this paper describes a "sub-classification" of the near-Antarctic class into even smaller groups. This sub-classification might be thought of as a two-level hierarchy, in that we are looking for classes within a class. As an extra benefit, focusing our clustering efforts on the near-Antarctic region means that the global data imbalance (i.e. the relative abundance of observations north of the PF) is less likely to bias our results.

The distribution of the near-Antarctic profiles used in the rest of this study features some spatial biases in coverage, which we handle carefully at the training stage (Fig. 2(a)). The earliest profiles are from CTD casts in the 1970s, although the vast





majority of the profiles are from the Argo float era, i.e. from the mid-2000s onward (Fig. 2(c)). The near-Antarctic profiles
are broadly characterised by a near-surface temperature inversion (i.e. colder at the surface and warmer at depth), which is

140 stabilised by the salinity stratification (i.e. fresher at the surface and saltier at depth). Such profiles are often described as "salt
stratified", indicating that their vertical stability depends on salinity and not temperature (Roquet et al., 2022). The widespread
presence of salt stratification in the near-Antarctic class is consistent with the property contrast usually seen across the Polar
Front, which acts as an approximate dividing line between waters bearing the imprints of near-Antarctic processes (e.g. ice shelf
melting and iceberg calving leading to a layer of near-surface freshwater) and waters bearing the imprints of more subtropical

processes.

## 2.2 Profile classification modelling

In this section, we use PCM to identify profile types within the near-Antarctic collection of profiles. First, we preprocess the
near-Antarctic profiles by standardising the temperature and salinity values on each pressure level. This ensures that, on each
pressure level, the temperature and salinity distributions have a mean of zero and a variance of 1.0. The advantage of scaling

each pressure level separately is that it allows all variations in temperature and salinity structure to be specified relative to the
observed variability on that pressure level. Without such standardisation, variations in the near-surface values, which tend to
have a strong seasonal imprint, would dominate the classification and obscure the structural importance of smaller variations
at depth.

After standardising the temperature and salinity values on each pressure level, we reduce the dimensionality of the dataset

by carrying out principal component analysis (PCA). This reduces the computational complexity of the classification task and
effectively filters out some of the small-scale vertical variability in the profiles. PCA may also be used to study ocean structures
by itself (e.g. Pauthenet et al. (2017)), but here we mainly use it as a dimensionality-reduction technique. We employ a six-
component PCA that retains roughly 95% of the vertical variability in the near-Antarctic profile dataset (Appendix B). PCA
reduces the number of "features" in the classification problem from 42 (i.e. temperature and salinity values on 21 pressure

levels) to six. It is within this six-dimensional abstract PC space that we wish to identify coherent sub-groups or classes.
However, this brings us to a complex issue in unsupervised classification: into how many different classes should we attempt
to sort the profiles?

Because oceanographic data is highly correlated across a variety of spatial and temporal scales, it cannot always be cleanly
separated into groups. As a result, there is not necessarily an objective measure for the "success" of any particular unsupervised

classification approach. Instead, techniques such as the PCM should be considered tools for exploration and discovery within
a dataset, allowing us to identify coherent regimes with similar vertical structures, and the boundaries between them, in a
probabilistic fashion. As with many classification problems, there is a tradeoff between the complexity of the classification
model and its interpretability. Simple two-class models are usually straightforward to interpret (e.g. colder waters versus
warmer waters), but they ignore more subtle structures in the data which may be of interest. As we increase the number of

170 classes into which we attempt to sort the data, we often lose interpretability as the classes often become increasingly difficult
to distinguish from one another. There are statistical tools that can help guide our choice of the number of classes, aiding us in



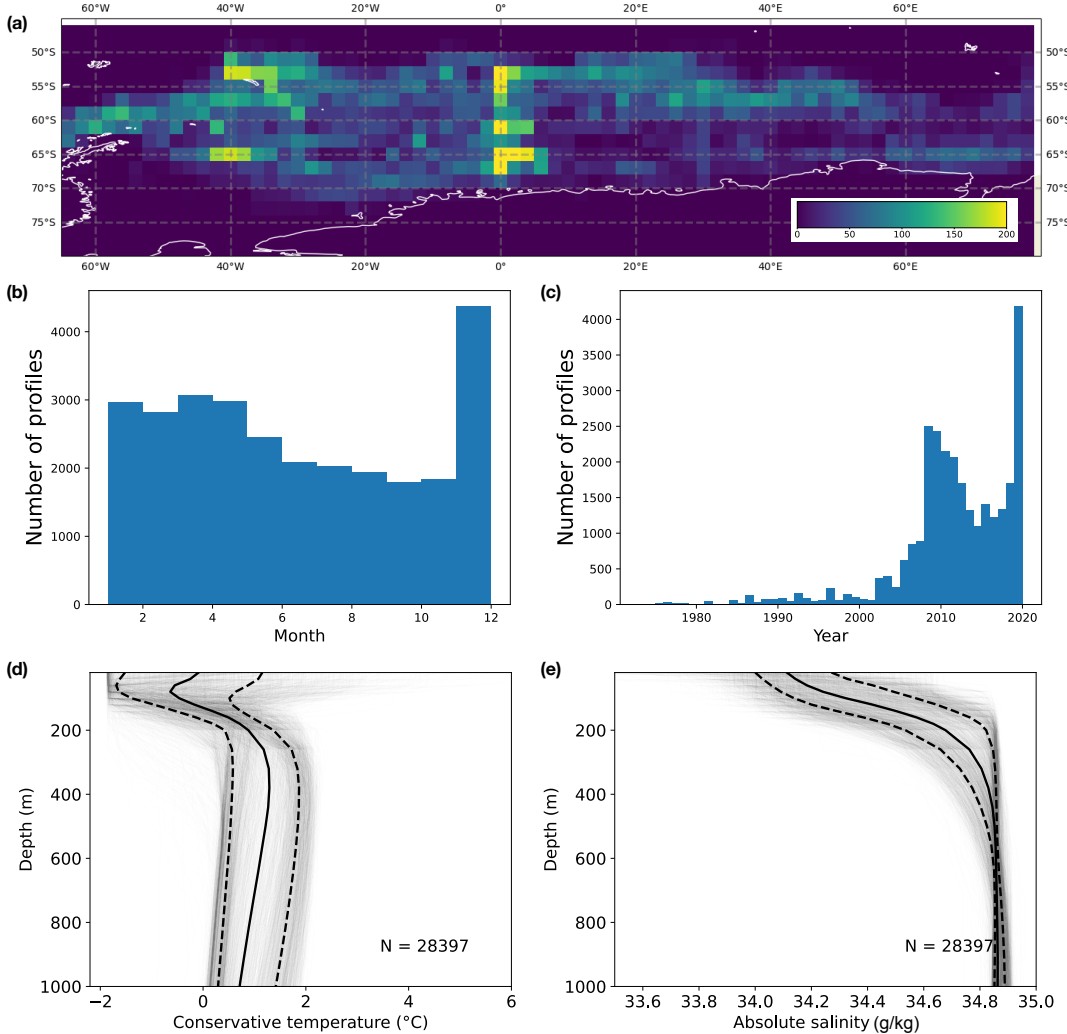

**Figure 2.** The distribution of profiles used in this study. (a) Profile locations in $2°$ latitude-longitude bins. (b) Distribution of the profiles by month. (c) Distribution of the profiles by year. (d) Random selection of 10% of the conservative temperature profiles by depth (thin grey lines), shown with the 25th percentile (thin dashed line, left), the median (solid black line), and the 75th percentile (thin dashed line, right). (e) Same as (d), for absolute salinity. The total number of profiles is $N = 28,397$.

avoiding the common pitfalls of underfitting and overfitting, but these tools often only provide a range, within which we are free to choose the level of complexity of the statistical model (Appendix B). This is roughly analogous to the "hierarchy of models" concept outlined by Held (2005), which suggests that we can learn a great deal about a system by studying what happens as we add or take away various sources of complexity. In the context of PCM, increasing the number of classes effectively increases the level of complexity of our representation of the system.





For this paper, we will focus on a four-class representation of the near-Antarctic profile dataset. This four-class model is simple enough to be readily interpretable, while being complex enough to identify differences between circumpolar waters, the transition between circumpolar waters and the gyre, the outer gyre, and the central gyre. Classification approaches must strike a balance between interpretability and complexity, and here we found the four-class model to be (1) within the constraints suggested by statistical criteria, and (2) sufficiently complex as to allow for a rich description of the system. Decreasing the number of classes to three loses the distinction between the gyre edge and the gyre core. Increasing the number of classes beyond four leads to an increasing amount of overlap between the classes, making them more difficult to interpret (Appendix B). In the next section, we describe the four-class description of the near-Antarctic profile dataset.

## 3 Results

In section 3.1, we describe the anatomy of the four-class PCM in terms of vertical temperature and salinity structure. For more details on the principal component expansion and the profile classification model, see appendix B. In section 3.2, we use a probabilistic classification metric and mixed layer depth to search for signatures of mixing. In section 3.3, we examine the classes for the characteristics of WW. In section 3.4, we identify the fingerprints of the core of CDW, as associated with a subsurface temperature maximum. In section 3.5, we explore the seasonal cycle of the classes and highlight the inferred characteristics of mixing between WW and CDW.

### 3.1 Structure of the classes

The four-class PCM identifies (1) a circumpolar class, (2) a transition class between the circumpolar waters and the Weddell Gyre, (3) waters on the edge of the gyre, and (4) waters in the core of the gyre. We can identify some key differences between the four classes by examining their vertical structures in temperature, salinity, and the resulting density (Figure 3). All descriptions here are intended to be relative between the four classes. The circumpolar class features a mean temperature distribution that is relatively warm and is characterised by a deep temperature minimum (median depth 120m, Fig. 3, top row). The layering observed in the top 1000m consists of alternating warm and cold layers, stabilised throughout by a gently-changing salt stratification with fresher water near the surface and saltier water at depth; the resulting density structure is well-stratified. When compared with the circumpolar class, the transition class, which sits between the more circumpolar waters and the more gyre-based waters, is characterised by a shallower temperature minimum (median depth 80m) and weaker salt stratification in the subsurface. The resulting density structure is slightly less stratified in the subsurface (Fig. 3, second row). The two gyre classes both feature a shallow temperature minimum and two salt regimes: specifically a region of increasing salinity with depth and a region of relatively uniform salinity. The resulting density structures reflect these salinity differences. The gyre edge features a more gradual change between these two salt regimes with depth, whereas the gyre core features a rapid change in salt stratification at around 200m depth (Fig. 3, bottom two rows). The median depth of the temperature minimum is 80m for the gyre edge class and 60m for the gyre core class, which is broadly consistent with the upwelling of CDW within the



cyclonic gyre centre. The temperature minimum, which indicates the presence of WW, deepens as we consider classes further away from the gyre core (Figure 3, from the bottom row to the top row).

**Figure 3.** Distribution of the conservative temperature, absolute salinity, and potential density anomaly ($\sigma_0$) of the four classes identified by the GMM algorithm. Each panel features a random 18% of the profiles (thin grey lines), the 25th percentile (dashed line, left), the median (solid line), the 75th percentile (dashed line, right), and the number of profiles $N$ in each class.

We find that the four classes occupy fairly distinct regions when plotted in longitude and latitude, with some expected overlap due to the highly correlated nature of ocean data and the probabilistic nature of our classification method (Figure 4). The circumpolar class is located broadly between the polar front (PF) and the southern boundary of the ACC (SBDY), although



some circumpolar-class profiles are found south of the SBDY between 20-40°E, a possible fingerprint of the conversion of ACC
waters to gyre-associated waters. The mean dynamic height anomaly of the 500m surface, chosen to give some indication of
the "height" structure of the classes, is deepest in the circumpolar class, with a median value of -3.0m. The transition class
straddles the SBDY in the western part of the domain and is more clearly south of the SBDY in the eastern part of the domain,
especially east of the Prime Meridian. As with the circumpolar class, this excursion south of the SBDY may be a fingerprint of
the conversion of ACC-sourced CDW to WDW (Fig. 1). It features shallower dynamic heights (median -2.3m), although the
excursion south of the SBDY has a height gradient. The gyre edge class features two distinct branches: the more western profiles
that sit just south of the SBDY and those found largely along $f/h$ contours near the Antarctic continent, where $f = 2\Omega\cos(\phi)$
is the Coriolis parameter ($\Omega$ is Earth's rotation rate and $\phi$ is latitude), and $h$ is the depth of the water column. The more
western pattern may be indicative of the Weddell-Scotia Confluence waters. When considered together, the two branches may
be thought of as largely aligning with the edge of the gyre, where its isopycnals dome down into the subsurface. Its median
dynamic height anomaly is -2.0m, although some regions in its western branch are shallower than the median. Finally, the gyre
core class is characterised by a dynamic height gradient from west to east, with an overall median value of -1.6m. Although
this PCM does not separate the western and eastern cells of the gyre circulation, the shape of the gyre core class reflects the
agglomeration of these two, for example in the northeast excursion between 0-20°E. As we consider the classes from north to
south (Fig. 4, top to bottom), the broad deep-to-shallow progression of the 500mb dynamic height surface is consistent with
broad upwelling in the gyre and downward-sloping surfaces further north.

## 3.2    Signatures of mixing

In this subsection, we use the distributions of the I-metric and the mixed layer depth to search for potential signatures of
mixing and transformation in the dataset. As discussed in section 2.2, all GMM-based classification methods, including the
ocean-specific PCM approach, are probabilistic. For each profile, the classification algorithm returns probabilities across the
classes. PCM assigns each profile to the class with the highest probability, but there is potentially useful information contained
in the distribution of probabilities across classes. In order to take advantage of this distribution, we plot the I-metric, which
uses the difference between the class with the highest probability and the class with the second-highest probability to quantify
the probability that a profile is located at a boundary between classes (Thomas et al., 2021). We can express the I-metric for a
single profile as:

$$I = 1 - \left[P(c = c_k)_{highest} - P(c = c_l)_{runner-up}\right], \tag{1}$$

where $P(c = c_k)_{highest}$ is the highest posterior probability that GMM has assigned to the profile, such that the profile has
been classified into class $c_k$, and $P(c = c_l)_{runner-up}$ is the second-highest posterior probability that GMM has assigned to
the profile. If $I$ is small, it indicates that the difference between the probabilities is large; profiles with small $I$ values are not
likely to be located near a boundary between classes. If $I$ is large, it indicates that the difference between the probabilities is
small; profiles with large $I$ are more likely to be located near a boundary between classes. With oceanographic profile data, the





**Figure 4.** Dynamic height of the 500 mb pressure surface in each of the four classes, binned and averaged over $1°$ latitude-longitude bins. Also shown are several fronts of the ACC from Kim and Orsi (2014), i.e. the Subantarctic Front (SAF), the Polar Front (PF), the southern ACC front (SACCF), and the southern boundary (SBDY). Also shown are contours of constant $f/h$ (thin grey lines), where $f$ is the Coriolis parameter and $h$ is the depth of the water column.

boundaries between classes may represent regions of mixing or transformation between profile types associated with changes in one or more water masses that make up the profiles under consideration.





The circumpolar class is characterised by low I-metric values across its entire distribution (median 0.0, third quartile $Q3 = 0.1$), except at some locations along its southern edge and in the excursion south of the SBDY between 20-40°E. Note that the low I-metric values along the northern edge of the circumpolar class should be interpreted in the context of the near-Antarctic dataset: there are no profiles north of the circumpolar class in this dataset, so the classification is unambiguous. The transition class features an I-metric distribution with the highest mean and third quartile across the four classes (median 0.1, $Q3 = 0.4$), consistent with its characterisation as a transition class between the more circumpolar waters and the more gyre-associated waters. The highest spatially coherent values, indicating high probability of being at or near a boundary, are found in the southward excursion between 0-20°E. The gyre edge class also features high I-metric values in some locations (median 0.0, $Q3 = 0.3$), particularly in the more western wing just south of the SBDY. The profiles near the Antarctic coast tend to have much lower I-metric values. Although it is difficult to state conclusively with profile data alone, the difference in I-metric values between the western wing south of the SBDY and the eastern wing along the Antarctic shelf is consistent with the gyre edge class including both inflowing waters at the eastern edge of the gyre and waters in its northern flank receiving input of shelf waters locally (Fig. 1). Finally, the gyre core class is characterised by low I-metric values (median 0.01, $Q3 = 0.04$), indicating an unambiguous "core" profile type, with some mixing and transformation along its edges. The slightly higher values between 0-20°E, co-located with the southward excursion of the transition waters, may indicate the signature of the transformation of CDW into WDW in the gyre core.

The mixed layer is a layer of relatively uniform density, also described as "weak stratification", in the upper ocean. It reflects the action of mixing processes that tend to homogenise the near-surface layer of the ocean. Here we examine the distribution of the mixed layer depth, in comparison with the I-metric, to look for potential signatures of regions where mixing has left an imprint on the profile structure. We use the integral depth-scale method described in Thomson and Fine (2003), which estimates the mixed layer depth $D$ as follows:

$$D = \frac{\int_0^{z_r} z N_b^2(z) dz}{\int_0^{z_r} N_b^2(z) dz},$$ (2)

where $z_r = 1000$m is an arbitrary reference depth and

$$N_b^2(z) = \left( -\frac{g}{\rho_0} \frac{d\rho_\theta}{dz} \right)^{1/2}$$ (3)

is the buoyancy frequency, $g$ is the acceleration due to gravity, and $\rho_0$ is a reference density. We calculate $N_b^2$ using the Gibbs seawater toolbox (McDougall and Barker, 2011).

Broadly speaking, the mixed layer is deeper in the more northern classes and shallower in the gyre core, with some local exceptions (Fig. 6). The circumpolar class features the deepest mixed layers (median 190m), with especially deep values between 20-40°E. The transition class, although shallower overall than the circumpolar class (median 140m), features deep mixed layers along the SBDY, indicating that the imprint of mixing is not uniform across the class. The gyre edge class (median 130m) also features regions of deeper mixed layers and shallower mixed layers, with deeper MLDs found along the SBDY and the eastern edge along the Antarctic continental shelf. Finally, the gyre core class has the shallowest mixed layers (median 100m), with some local exceptions along the Antarctic-adjacent $f/h$ contours.





**Figure 5.** Inter-class comparison metric (i.e. the I-metric) binned and averaged over 1° latitude-longitude bins. The metric can be interpreted as the probability that the profile is located at a boundary between classes within the Weddell Gyre dataset. The I-metric is defined in Thomas et al. (2021). The fronts and $f/h$ contours are the same as in Figure 4.





**Figure 6.** Mixed layer depth binned and averaged over $1°$ latitude-longitude bins. Mixed layer is calculated using an integral approach (Thomson and Fine, 2003). The fronts and $f/h$ contours are the same as in Figure 4.

## 3.3 Signatures of Winter Water

As discussed in section 3.1, the subsurface temperature minimum seen across the profiles is associated with the WW layer that is formed and renewed seasonally and transported by the near-surface currents. In a broad sense, the depth of the temperature



minimum is shallowest at high latitudes and deepest at lower latitudes, although there are localised differences (Fig. 7). In the circumpolar class, the minimum temperature layer is deeper along the northern extent and shallower along its southern extent,

and there is a corresponding gradient from warmer minimum values to colder minimum values from north to south (domain-wide median 0.16°C). In the transition class, the temperature minimum is especially deep along the SBDY in the western edge, roughly between the gyre and the Scotia Sea. This is an expected characteristic of the Weddell-Scotia Confluence waters, and also of enhanced deep winter convection associated with cold shelf waters (Whitworth et al., 1994). The minimum temperature values there are somewhat warmer than those in the rest of the transition class, especially compared with the cold southward

excursion between 0-20°C. The domain-wide median value of the minimum temperature is colder than that of the circumpolar class (median -0.9°C). The gyre edge and gyre core classes feature extremely cold minimum temperature values (gyre edge median -1.7°C, gyre core median -1.8°C), and the gyre core distribution is colder overall (gyre edge $Q3 = -1.3$°C, gyre core $Q3 = -1.6$°C). Locally in the gyre edge class, as with the transition class, the depth of the temperature minimum is especially deep along the SBDY on the western edge of the domain, although the temperatures are slightly warmer there. The gyre core

class features an exceptionally shallow and extremely cold near-surface temperature minimum layer, with a slightly warmer and deeper edge. The processes that establish, renew and transport the WW layer all leave their imprints on the spatial distributions of the temperature minima and the depths at which those minima are found.

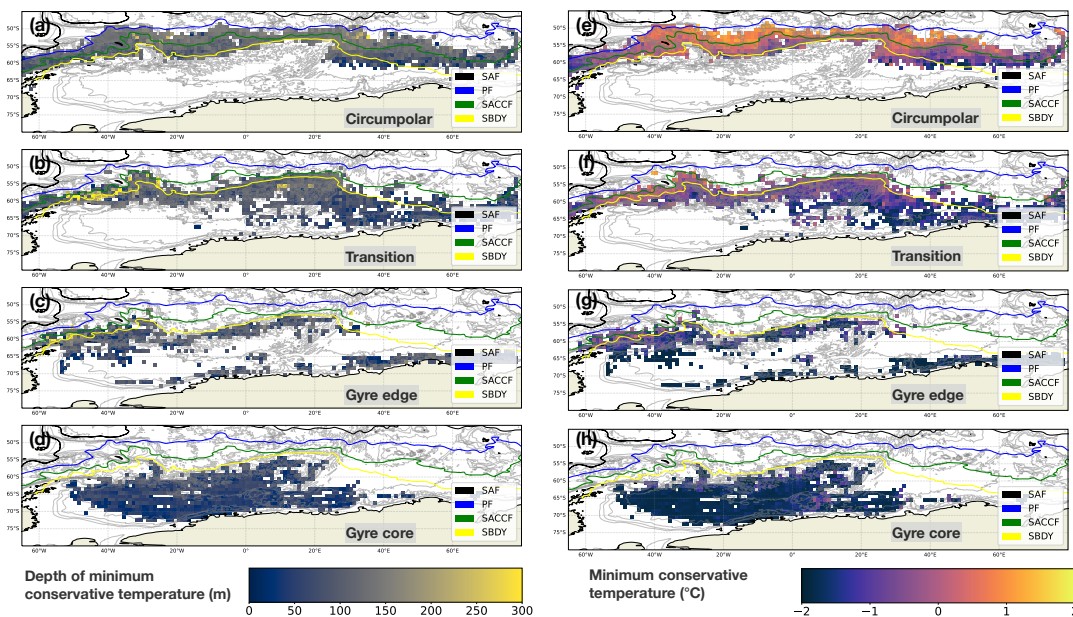

**Figure 7.** Depth of the conservative temperature minimum (left column) and the value of the conservative temperature minimum (right column), binned and averaged over 1° latitude-longitude bins. The fronts and $f/h$ contours are the same as in Figure 4.



## 3.4 Signatures of Circumpolar Deep Water

Most of the profiles in the near-Antarctic dataset feature a subsurface temperature maximum. The temperature maximum
represents the core of CDW; we can use this layer to track the core of the CDW as it is transported from the circumpolar region
to the Weddell Gyre (Reeve et al., 2016, 2019). In the circumpolar class, we see the core of the CDW deepen and cool along the
length of the ACC, with especially deep and cold values in the southward excursion south of the SBDY between 20-40°E (Fig.
8). In the transition class, we see a gradient in the CDW-associated temperature maximum, getting colder along the southward
excursion, where it meets up with the western edge of the gyre core and the WDW. The gyre edge class features particularly
deep and cold temperature maxima (median $T_{max} = 0.7°$C, median depth 500m), with the deepest $T_{max}$ depths of all the
classes. By contrast, the gyre core class is characterised by somewhat shallower temperature maxima (median $T_{max} = 0.7°$C,
median depth 260m The fact that the gyre edge class exhibits a much deeper temperature maximum than the gyre core class is
a key difference between the two, possibly highlighting that the processes setting the depth of $T_{max}$ may be different between
them. The gyre edge class may be more associated with the circulation of WSDW around the edge of the gyre and the doming
down of isopycnals along its boundary (Fig. 1). Notably, there is a clear difference in the depth of the temperature maximum
along the gyre core in the east-west direction, with a deeper layer in the west and a shallower layer in the east (Fig. 8(d)). This
is consistent with CDW intrusion into the eastern edge of the gyre class.

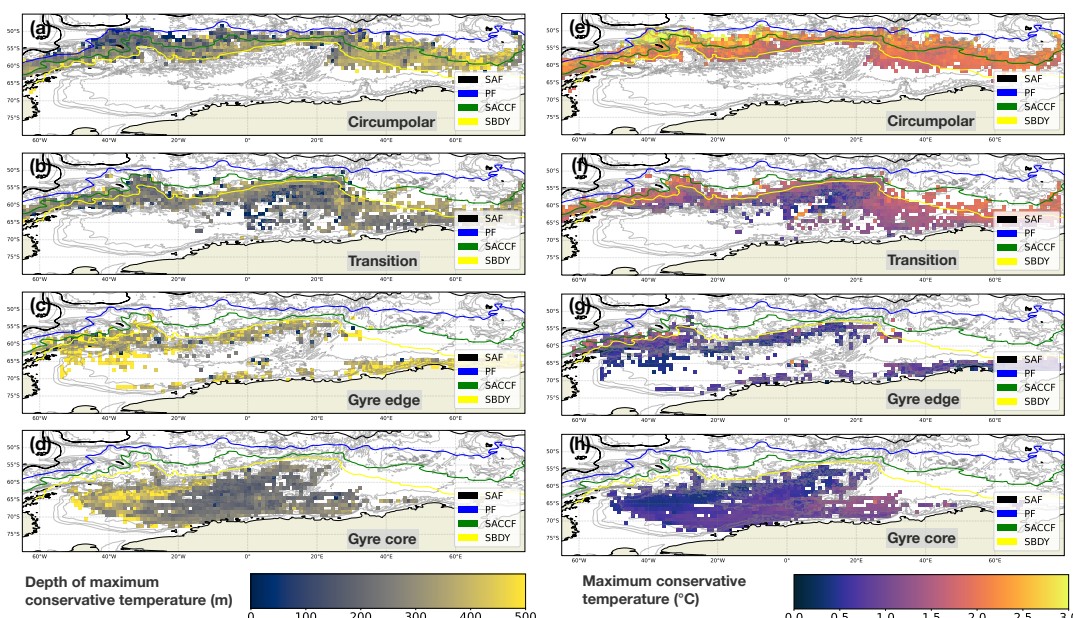

**Figure 8.** The depth of the conservative temperature maximum (left column) and the value of the conservative temperature maximum (right
column), binned and averaged over $1°$ latitude-longitude bins. The fronts and $f/h$ contours are the same as in Figure 4.



## 3.5 Signatures of mixing between Circumpolar Deep Water and Winter Water

All four classes are characterised by a proportion of relatively salty, dense and warm water, and a proportion of relatively fresh,
cold and light water (Fig. 9). The salty, dense, warm waters have properties consistent with CDW, and occupy the right-hand
side of the T-S plots in Fig. 9. The fresh, cold, light waters have properties consistent with a layer of temperature minimum
WW, sitting along the bottom of the T-S plots. The WW becomes increasingly salty as we consider the classes in order of
proximity to the gyre core: i.e. circumpolar, transition, edge, core. Broadly speaking, there are two possible causes for this: (1)
the effect of brine rejection associated with sea ice formation leads to saltier WW varieties, and (2) the core of CDW, which is
characterised by a subsurface temperature maximum, is shallower in the gyre classes due to upwelling. Both of these signatures
lead to enhanced vertical mixing between CDW and the overlying WW.

Although all four classes feature CDW, WW and surface waters (SW), there is a key difference between the circumpolar
waters and those of the transition class: compared with the circumpolar class, the transition class is relatively more affected
by brine rejection from sea ice formation and the mixing associated with this brine rejection. In particular, there is a range of
T-S space that is unoccupied in the circumpolar class, but relatively more populated in the other classes, notably the transition
class (Fig. 9, grey oval). We hypothesise that this region in T-S space is associated with mixing between CDW and WW; the
circumpolar class is relatively less affected by this mixing, whereas the transition class is relatively more influenced by this
mixing. Furthermore, the two gyre classes feature strong mixing between CDW and WW, enabled in part by upwelling in the
cyclonic gyre. The "mixing line" between CDW and WW is especially compact and straight in the gyre core class, which is
consistent with CDW and WW being physically closest to one another there.

Across all four classes, the seasonal cycle is characterised by a weakening of the near-surface temperature stratification
in austral winter and spring (JJA and SON), followed by restratification in the austral summer and autumn (DJF and MAM)
as the SW warm (Fig. 10). The layer of WW remains trapped under the warmed, restratified near-surface waters, decaying
somewhat by mixing until it is ventilated again during the following winter and spring. The relatively warmer waters of the
CDW that sit below the WW remain steady throughout the year, as they are relatively isolated from the processes associated
with the seasonal cycle. In terms of salinity, all four classes feature year-long stability in terms of fresher waters overlying
saltier waters, although there is a clear near-surface seasonal cycle (not shown).

The circumpolar and transition classes are characterised by the formation of WW in winter and spring, with its rapid erosion
in summer and autumn (Fig. 11, top two rows). The circumpolar class features a strong seasonal cycle in SW, warming and
cooling largely by heat transfer with the atmosphere. The transition class also displays a relatively strong seasonal cycle in
SW. In contrast, in the gyre edge and gyre core classes, the WW is present year-round, and mixing between WW and CDW, as
inferred by the density of profiles along the "mixing line" between the two water masses, appears to be weakest in winter and
stronger throughout the rest of the year.



**Figure 9.** Temperature-salinity diagrams for the four classes. The shading indicates the mean I-metric value for each bin (0.1°C, 0.025 g/kg). Since each profile has a single I-metric value, the approach used here averages all profiles that pass through any given bin. The result is a relative indication of the core class properties versus the more peripheral ones. The solid lines are potential density contours ($\sigma_0$). The grey oval indicates a potential signature of mixing between the winter water and deeper water. Also shown are the approximate locations of the surface water (SW), circumpolar deep water (CDW), and winter water (WW).





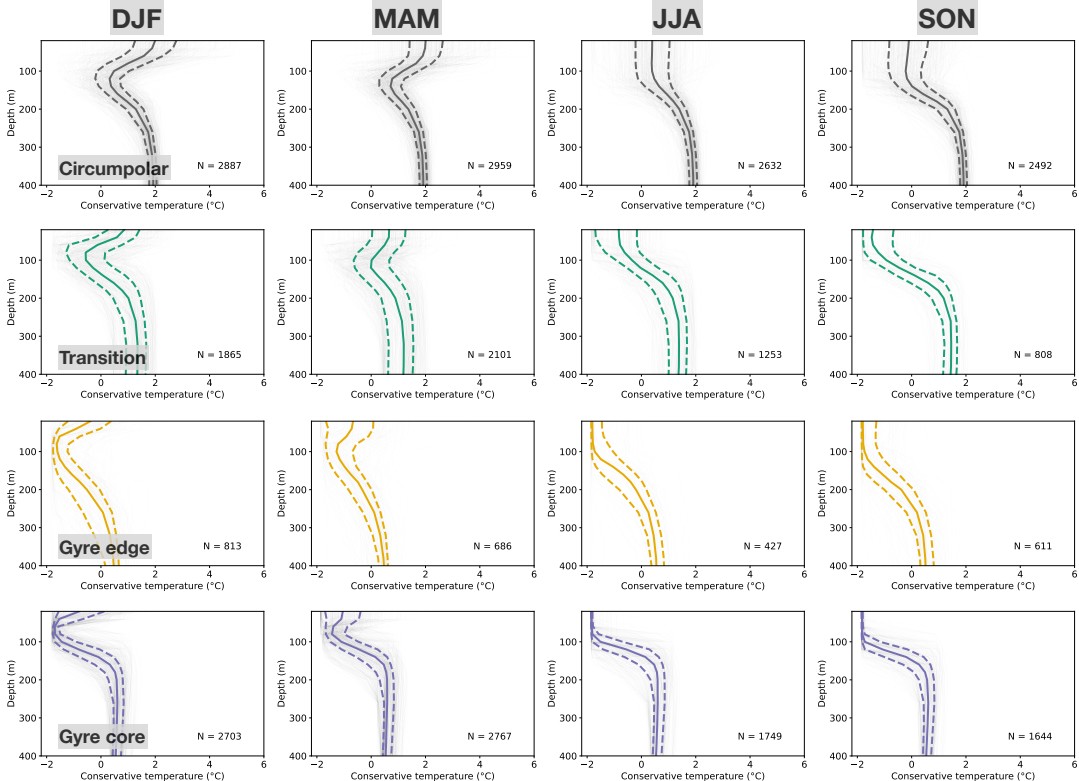

**Figure 10.** Distribution of the conservative temperature of the four classes, split by season. Each panel features a random 18% of the profiles (thin grey lines), the 25th percentile (dashed line, left), the median (solid line), the 75th percentile (dashed line, right), and the number of profiles $N$ in each class and season.

## 3.6 Physical context: surface stress, upwelling, and sea ice

Here we discuss two physical fields that are relevant to our interpretation of the classes: surface stress-driven (Ekman) upwelling, and freshwater flux from brine rejection. In the Southern Ocean, surface stress is set by near-surface winds and modulated by the presence of sea ice, through both ice coverage of the ocean surface and stress associated with sea ice drift (Dotto et al., 2018). The Ekman upwelling velocity at the base of the Ekman layer is

$$w_{ek} = \frac{1}{\rho_0} \left[ \nabla \times \left( \frac{\tau}{f} \right) \right], \tag{4}$$

where $\rho_0$ is the reference density, $\tau$ is the surface stress from ERA-5 reanalysis following Dotto et al. (2018), and $f$ is the Coriolis parameter. We find climatological upwelling of various intensities along the SBDY, which is co-located with the edge of the circumpolar class and the core of the transition class (Fig. 12(a)). There is also weaker upwelling throughout the gyre core, which is consistent with its cyclonic circulation and the associated upwelling of CDW throughout. We find downwelling in the western edge of the domain near the Filchner-Ronne Ice Shelf, and patches of strong downwelling along the rest of the





**Figure 11.** Temperature-salinity diagrams grouped by class and season. The color indicates the number of profiles passing through each T-S bin (width 0.1°C, 0.025 g/kg), and the solid lines are potential density contours.

Antarctic coast (Fig. 1). This near-coastal downwelling seemingly affects the southern extent of the gyre edge class as well as the edges of the gyre core class.

    In order to further investigate our hypothesis that the transition class is relatively more influenced by brine rejection from sea ice formation compared with the circumpolar class, we examine an estimate of salt input associated with winter sea ice



formation from a regional state estimate (Mazloff et al., 2010). We find that in the western part of the domain, the strongest
brine rejection largely coincides with the position of the SBDY, which again is co-located with the core of the transition class
(Fig. 12(b)). This positioning is consistent with the hypothesis that brine rejection is able to affect the transition class more than
the circumpolar class, which is found mostly north of the SBDY. In the eastern part of the domain, the salt input associated with
sea ice mostly aligns with the SACCF, but this is somewhat downstream of our main analysis region near the Weddell Gyre.
These results are compatible with those of Haumann et al. (2016), who find both local freezing and transport to be important
for setting the freshwater input south of the region of maximum sea ice extent (their Fig. 4).

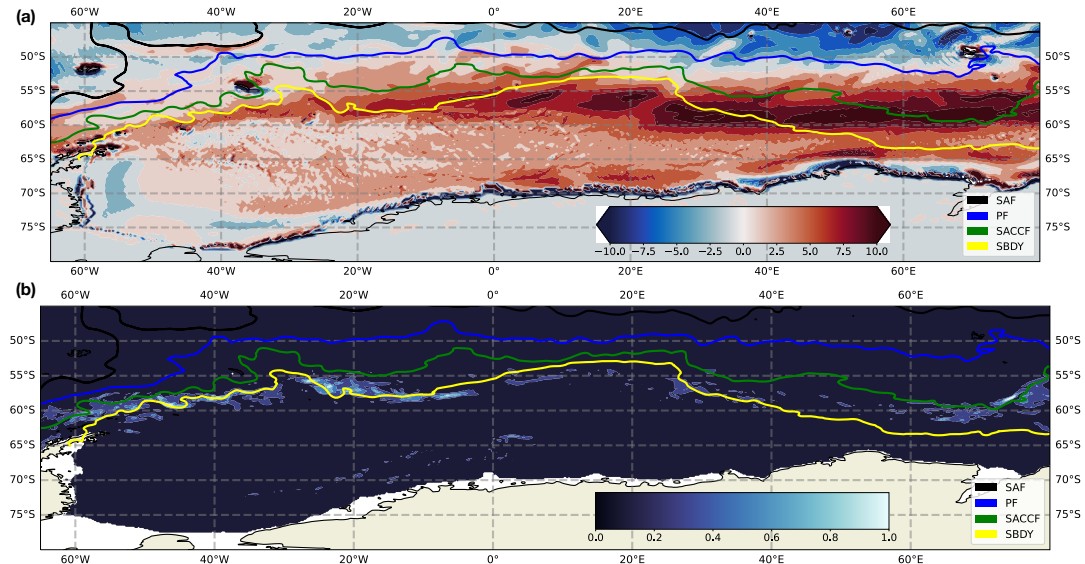

**Figure 12.** (a) Estimated upwelling ($10^{-4}$ m s$^{-1}$) from surface stress, including the effects of wind stress, sea ice coverage, and sea ice
drift, following Dotto et al. (2018). Sea ice concentration is from Climate Data Record v4 and sea ice drift is from Polar Pathfinder. (b)
Approximate position of the wintertime sea ice freezing line, as indicated by the salt input from brine rejection during sea ice formation
(negative freshwater input, shown in units of $10^{-5}$ kg m$^{-2}$ s$^{-1}$). Values below $10^{-1}$ kg m$^{-2}$ s$^{-1}$ have been masked out (i.e. set to zero).
Data from SOSE averaged over JAS during the period 2005-2010 (Mazloff et al., 2010). The fronts are the same as in Figure 4 from Kim
and Orsi (2014).

## 4   Discussion

The Weddell Gyre region controls the properties of some of the densest waters on the planet, with associated effects on
the overturning circulation and on the air-sea partitioning of heat and carbon (Vernet et al., 2019). Because of these unique
properties, the structure and variability of the Weddell Gyre are relevant to how it functions as part of the global ocean and
climate system. In this work, we applied a profile classification model (PCM), which is a type of unsupervised machine
learning, to a temperature and salinity profile dataset covering the Atlantic sector and part of the Indian sector of the Southern





Ocean. Our objective was to use the PCM as a "discovery tool" to identify both expected and underappreciated thermohaline structures in the Weddell Gyre region and to examine their seasonal variability. Our application of PCM identified four coherent "profile types" that can be described as follows: (1) a circumpolar class that largely sits north of the SBDY, (2) a transition
class between the circumpolar region and the gyre region, (3) a gyre edge class with northern and southern regions, and (4) a gyre core class. The four classes are reasonably interpretable and geographically distinct from one another, with some overlap as expected when using a probabilistic classification scheme such as PCM on a highly-correlated dataset.

The unsupervised classification approach used here identifies, in an objective, data-driven way, some key features of the Weddell Gyre and its circulation. Some of these features are already well-understood and characterised from somewhat more
subjective, expertise-driven interpretations of water mass properties (e.g. large-scale upwelling in the gyre interior and down-welling near the Antarctic margins), but other features are not as well-documented. For instance, PCM identifies the inflow of CDW across the eastern and northern boundaries of the Weddell Gyre, the extent of the Weddell-Scotia Confluence, and the spatial distribution of mixing between WW and CDW. It also characterises the seasonal cycle of the four classes, highlighting the formation, extent, and destruction of WW throughout the gyre and its surroundings, via mixing with the underlying CDW.

**4.1 Sea ice processes and class structures**

Sea ice growth, brine rejection, and vertical mixing are closely related in the polar regions (Gordon, 1990; Martinson, 1990). A key result from this study is the marked difference between waters where WW and CDW more readily mix with each other (i.e. the transition and gyre classes) and the more northern class where the WW and CDW are relatively more isolated from one another (i.e. the circumpolar class); this is especially apparent when viewed in T-S space (Fig. 11). In particular, we hypothesise
that the structure of the transition class is affected by salt input from sea ice processes (e.g. brine rejection) and the associated mixing, whereas the circumpolar class is not especially affected by this process. Local freezing and sea ice formation tend to add salt to the regions occupied by the transition class in a climatological sense, which would weaken the stratification and thereby encourage mixing between WW and CDW (Haumann et al., 2016). This process is most relevant in JJA and SON, when sea ice reaches its maximum extent and is most aligned with the SBDY in the western Weddell Gyre (Fig. 12). Although
this same region is also affected by freshwater flux due to melting and sea ice export, which would have a stabilising effect on the water column during some parts of the year, this would not necessarily eliminate the effect of wintertime salt flux from brine rejection and the relative increase in mixing.

As an alternative hypothesis, it may be that the weak stratification of the transition class is related to the spillage and subsequent eastward advection of relatively dense and weakly stratified shelf waters from the tip of the Antarctic Peninsula.
Those waters have had their stratification reduced by sea ice production and brine rejection on the continental shelves of the western Weddell Sea. Quantifying the relative effect of local brine rejection and advection on the transition and circumpolar classes may be a fruitful avenue for future observational and numerical studies.





## 4.2 Influences of bathymetric and shelf processes on class structures

Using PCM, the boundaries between classes are probabilistic and somewhat "fuzzy". That being said, many of the approximate
class boundaries roughly coincide with Southern Ocean fronts, the positions of which are strongly linked to bathymetry. As an
example, the boundary of the circumpolar class approximately aligns with the SBDY, which is constrained by the South Scotia
Ridge. This correspondence is consistent with studies that have shown the importance of bathymetry, including ridge geometry,
in determining the stratification, circulation, and extent of the Weddell Gyre (Patmore et al., 2019; Wilson et al., 2022).

Processes on the continental shelves also affect the structure of the classes. For example, ventilation may affect the structure
of the southern wing of the gyre edge class, as it is largely distributed along the Antarctic continental shelf. The transition
class may be affected by ventilation by low-salinity shelf waters along the eastern Weddell Gyre. These shelf waters are not
dense enough to support AABW formation; instead, they cascade down along the continental slope and ventilate the CDW.
(Thompson et al., 2018).

## 4.3 Weddell-Scotia Confluence waters

The Weddell-Scotia Confluence is a region of reduced mid-layer stratification that is located near the South Scotia Ridge
(Patterson and Sievers, 1980). Although this reduced stratification was initially thought to be a consequence of vertical mixing
caused by topography, it is now understood that the reduced stratification is due to shelf water spilling off the tip of the Antarctic
Peninsula and spreading eastwards (Whitworth et al., 1994). While each class or profile type in a PCM will necessarily bear
the imprint of many different processes, the northern branch of the gyre edge class appears to be spatially coincident with the
Weddell-Scotia Confluence (Fig. 5). Along the length of the SBDY, proceeding from west to east, the I-metric values of the
gyre edge class profiles tend to increase, indicating some mixing between profile types. PCM is not able to distinguish causes
or dynamical connections; however, it is notable that the northern branch remains coherent along the length of most of the
SBDY, where the Weddell-Scotia Confluence waters are thought to lie.

## 4.4 Northern and southern extents of the gyre edge class

The gyre edge class features both northern and southern branches. These two classes do not separate if the number of classes is
increased; their structure is similar enough to each other as to warrant grouping together by the PCM. Still, there are systematic
differences between the northern and southern branches (Fig. 13). The northern branch features warmer, slightly fresher, and
slightly lighter surface waters compared with the southern branch. Although we should be careful not to over-interpret a
possible dynamical connection between the branches, this pattern is broadly consistent with the warming and freshening of
surface waters as they flow along the ASF, mixing with the fresh shelf waters and warming as they turn northwards to join
the WSDW. By contrast, the northern branch features slightly colder, saltier, and denser subsurface waters below about 100m
depth. Overall, the change in stratification with depth is more gradual in the southern class and more abrupt in the northern class.
Again, although PCM groups these two regions together, one could reasonably decide to "manually" split these two branches





into separate classes for some applications. Further study on a possible dynamical connection between the two branches would
be welcome.

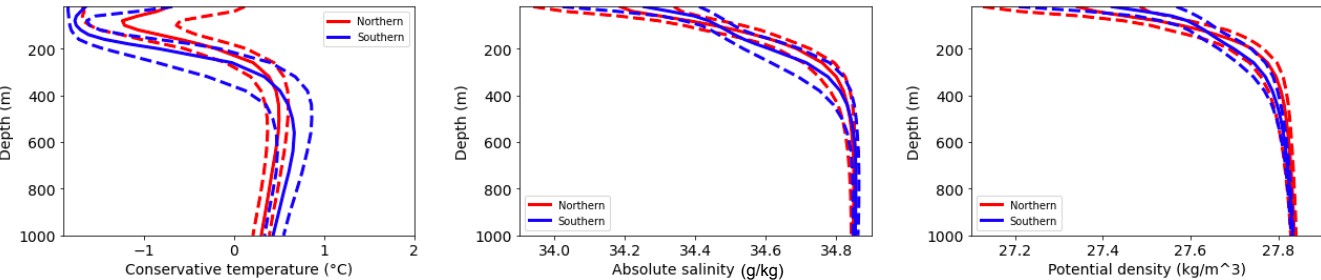

**Figure 13.** Compare northern and southern extends of the gyre edge class

## 4.5 The case for regional, application-specific profile classification models

As with many classification problems, there is a balance to strike between the interpretability of the PCM and its accuracy,
i.e. its ability to represent the full underlying covariance as captured by the available data. The ideal balance depends on the
objective of the study; if interpretability is the main objective, then one can opt for fewer classes, with less overlap between
them. If accuracy is the main objective, then one can opt for more classes, with more overlap between them. Typically, as
overlap increases, the ease of interpretation decreases. That being said, even in the simplest case with as few as two classes,
each "profile type" will bear the integrated signatures of many different processes (e.g. mixing between water masses, air-
sea interactions), so they must be interpreted in a way that considers this integration. Some statistical guidance is available
for helping with our choice of the number of classes, but this guidance often returns a range of allowed numbers of classes,
as opposed to a single objective value (Fig. B3). It is sometimes instructive to compare the structures of PCMs with different
numbers of classes, much as one might compare different levels of complexity in a hierarchy of numerical models (Held, 2005).
One can learn about the structure of the system by observing what changes as one adds or removes sources of complexity, in
this case by adding or removing classes.

The structure of the PCM also depends on its temporal and spatial domain. The goal of identifying a single global, unified
PCM is likely an impractical one, as the "noise" in a global PCM might well be meaningful "signal" in a more regional
PCM. This is analogous to identifying fronts in the Southern Ocean; the way we treat structures and the boundaries between
them benefits from a more region-specific, application-specific approach, for similar reasons - one application's "signal" is
another application's "noise" (Chapman et al., 2020; Thomas et al., 2021). As with many atmospheric and oceanographic
problems, filtering, signal processing, and the context of the variability are all routinely used to highlight certain features and
aid understanding. The flexibility of PCMs is one of their strengths; they can be focused on specific regions, variables, and
types of variability depending on the objective of the application at hand.



That said, the four-class model derived here could be compared to the Southern Ocean temperature-only PCM presented in Jones et al. (2019). In that study, there are only two classes south of the Polar Front, i.e. a near-Antarctic class located broadly in the gyres and along the slope current, and a more circumpolar class that runs just south of the Polar Front. As we found in the Weddell Gyre PCM, the profile types found south of the PF are characterised by salt stratification. Although one should be careful not to over-interpret, the circumpolar class in the temperature-only Southern Ocean PCM is roughly analogous to the circumpolar class found here in the Weddell Gyre-specific PCM. The Weddell Gyre PCM is able to distinguish additional structure because it (1) includes salinity as a variable, and (2) is focused on a regional set of profiles.

### 4.6 Identifying the underlying covariance structure

The actual grouping or classification of the profile data happens in an abstract principal component space. In PC space, the covariance structure of the dataset, both in terms of its climatology and the seasonal cycle, is represented by a "circumpolar wing" and a "gyre wing", separated by a transition bridge between the two. This is similar to what one sees using t-distributed Stochastic Neighbour Embedding (t-SNE), which is a non-linear dimensionality reduction technique that can be useful for visualising high-dimensional data (Fig. 14).

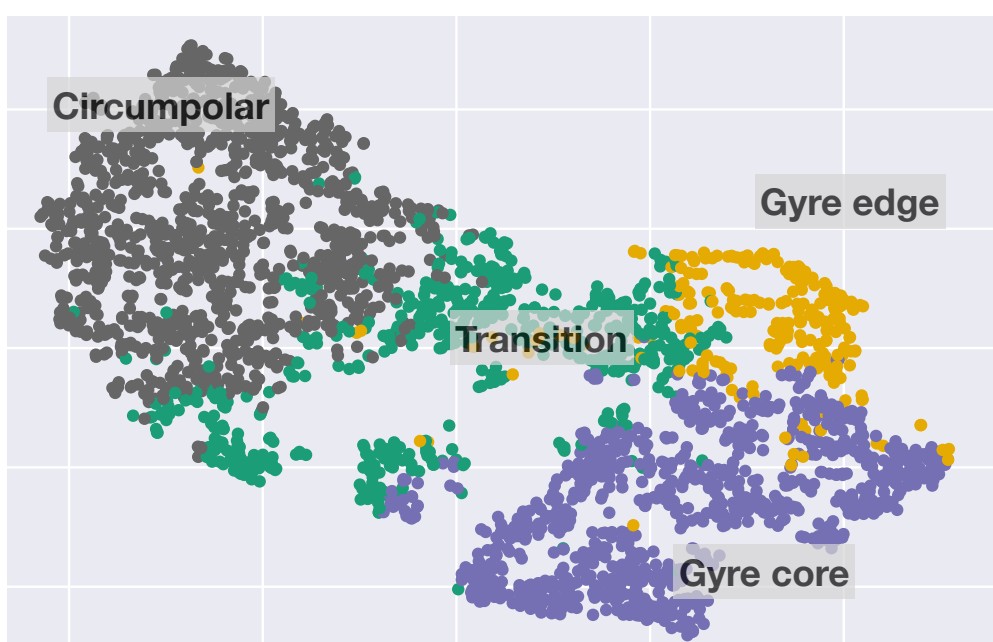

**Figure 14.** Visualisation of six-dimensional principal component space, transformed into a 2D space using t-SNE (Maaten and Hinton, 2008). The colours indicate the four classes into which the profiles have been sorted. The axes are arbitrary dimensions and should not be interpreted quantitatively; they are for visualisation purposes only.





 In this abstract space, one might consider the probability of a transitioning from the circumpolar wing to the gyre wing via the transition class. This framework makes the most sense in a more barotropic conceptual model of the Weddell Gyre, which may be defensible in the upper O(1000m) of the region, since the circulation in this layer is more likely to be equivalent barotropic compared with the entire water column (Killworth, 1992; Marshall, 1995; Krupitsky et al., 1996). The presence of baroclinicity implies the presence of shear, which likely only happens in specific locations (e.g. associated with bathymetric

features). Another way to frame this would be to ask - what transformations would be necessary to convert a profile from the circumpolar wing to the gyre wing, via the transition bridge? Examining the profile types, we can speculate that such a transformation would involve cooling throughout the water column, a shoaling and weak salinification of the WW layer, a deepening of the core of the CDW, and an overall weakening of the subsurface stratification.

### 4.7   Spatial and temporal coverage

The set of profiles used in this study are not distributed uniformly in space or season. Some locations are over-represented, for instance along repeat hydrographic sections (e.g., A12, A13.5). In order to minimise the effect of this spatial bias on the PCM, we selected training datasets that are approximately uniform in terms of areal coverage. We also carried out a sensitivity test wherein the training datasets were generated randomly with no guarantee of uniformity, and the results were broadly unchanged (not shown). In terms of temporal coverage, the austral winter and spring months are somewhat under-represented relative to

the rest of the year (Fig. 2(b)), which may bias the PCM towards summer and spring structures. However, since the total number of profiles in any given class and season is fairly small, we decided to use all available data instead of sub-selecting further for the training dataset. Although this does introduce what we expect to be a small bias in the PCM, we weighted this against the size of our dataset and decided to proceed with more profiles rather than fewer. The profile dataset is also non-uniform by year; this will also introduce some biases into our PCM. Because we are basically using the PCM to build up

a climatological and seasonal mean picture of the structure of the Weddell Gyre, we do not think that our results would change drastically using uniform sampling by year. Nevertheless, this bias highlights the need for continued profile-based monitoring of the Weddell Gyre and the surrounding areas, so that we may develop a more complete understanding of the seasonal cycle as well as interannual-to-decadal variability.

### 5   Conclusions

Through the processes of dense water formation and upwelling that occur in and around the Weddell Gyre, this unique region connects the atmosphere, surface ocean, and the deep ocean. Its structure and variability are relevant to how it functions as part of the global ocean and climate system. In this work, we used a profile classification model (PCM), a type of unsupervised classification built on Gaussian mixture modelling (GMM), as a tool for characterising temperature and salinity profile data in the Weddell Gyre region. The PCM highlights both expected and underappreciated structures, helping to generate new

hypotheses for further investigation. Specifically, the Weddell Gyre PCM identified four different "profile types", namely (1) a circumpolar class, (2) a transition class between the circumpolar profiles and the gyre profiles, (3) a class of profile on



the edge of the gyre, and (4) a class of profiles in the core of the gyre. The vertical structure and geographic distribution of the classes feature signatures of CDW inflow across the eastern boundary of the Weddell Gyre, the presence of the Weddell-Scotia Confluence, and the spatial distribution of mixing between the WW and CDW. The seasonal cycle of the class structure

highlights the formation, extent, and mixing of WW throughout the gyre and its surroundings. We put forward the hypothesis that the transition class is relatively more affected by mixing encouraged by brine rejection from sea ice formation compared with the circumpolar class, as suggested by the spatial overlap between the northern edge of winter sea ice production and the transition class. Unsupervised classification approaches such as PCM can complement existing expertise-driven analysis, which has been and will remain useful well into the future. Future studies that examine alternative classification strategies (e.g.

agglomerative clustering) would be a welcome addition to the literature.

*Code and data availability.* The dataset used in this study consists of profiles taken by Argo floats (http://argo.ucsd.edu) and ship-based CTDs as recorded in the World Ocean Database (https://www.ncei.noaa.gov/products/world-ocean-database). SOSE Iteration 100 data available from Scripps Institution of Oceanography (http://sose.ucsd.edu/sose_stateestimation_data_05to10.html). CDRv4 available via NSIDC (https://nsidc.org/data/G02202), and Polar Pathfinder sea ice drift data is also available via NSIDC (https://nsidc.org/data/nsidc-0116). The

code used to perform this analysis and produce the figures used in the paper is available via Zenodo (Jones, 2022). The full dataset used in this work is also available via Zenodo (Jones and Zhou, 2022).

## Appendix A:  Five-class PCM of entire South Atlantic Ocean and Indian Ocean dataset

The initial set of profiles used in this study covers the South Atlantic and part of the Indian Ocean (section 2.1). As a first classification step, we applied the PCM technique to this dataset in order to identify a coherent set of near-Antarctic profiles.

We chose to use this data-driven approach for consistency with the rest of our analysis, although in practice selecting all profiles south of the Polar Front would not substantially change our results, especially not those closer to the Weddell Gyre. Starting with the cleaned and prepared South Atlantic and Indian Ocean data, we followed the PCM procedure described in section 2. First, we reduced the dimensionality of the data using a six-component PCA, retaining 95% of the variability. We used the elbow method with both BIC and AIC to estimate the optimal number of classes, which in this case was $K = 5$; at $K = 5$, the

slope of both the mean BIC and AIC curves changes considerably, and BIC does not change much for $K > 5$.

The five-component PCM consists of these profile types: (1) subtropical Atlantic, (2) subtropical Indian, (3) circumpolar (more northern), (4) circumpolar (more southern), and (5) near-Antarctic (Fig. A1). There are some excursions of the circumpolar (southern) class south of the Polar Front, especially in the easternmost part of the domain, but the near-Antarctic class mostly consists of those profiles that are located south of the PF. The main analysis in this paper is a sub-classification of the

"near-Antarctic" profiles.



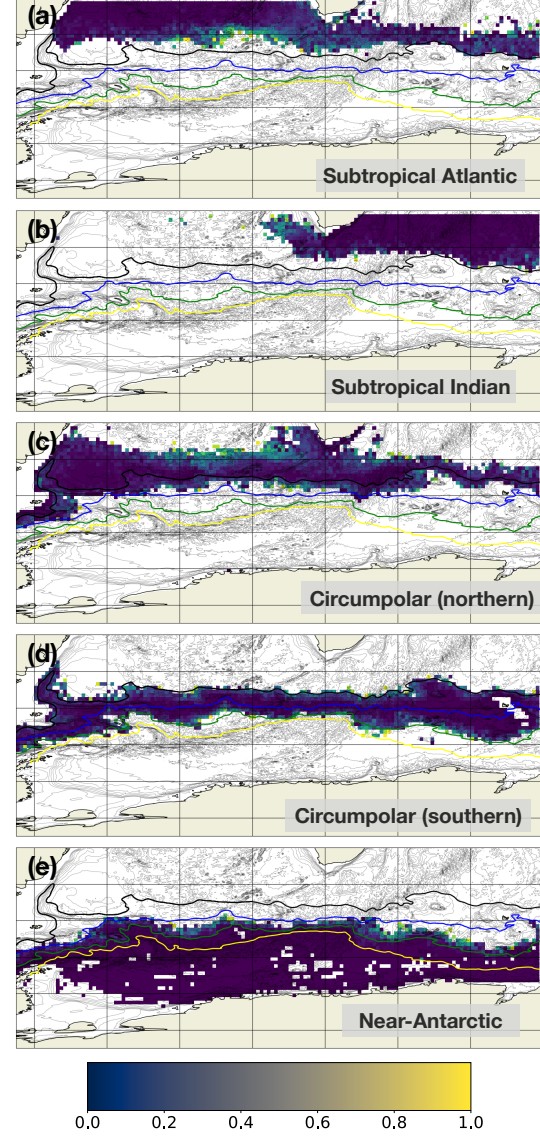

**Figure A1.** A five-class PCM of the initial dataset. The quantity shown in the I-metric, which is the probability that a profile is near a boundary between classes, as described in section 3.2. Also shown are fronts of the ACC, as described in Fig. 5. The rest of this study is essentially a "sub-classification" of the near-Antarctic class.

## Appendix B: Details of PCA and PCM

In this appendix, we describe some details of the PCA and PCM approaches used in this work. In section B1, we discuss dimension reduction using the principal component approach. In section B2, we describe the procedure used to select the num-




ber of classes. Finally, in section B3, we provide some mathematical details for GMM, which is the underlying unsupervised
classification approach used in our PCM.

## B1 Dimension reduction using principal component analysis

Principal component analysis (PCA) attempts to identify a set of eigenfunctions that can efficiently represent a dataset, specifically by representing its variability as a linear combination of these eigenfunctions. In this analysis, we use PCA to represent the vertical variability in the dataset by identifying a set of eigenvectors (i.e. principal components or PCs) that are functions
of depth. We find that a six-component representation of the variability in both conservative temperature and absolute salinity retains 95% of the variability within the dataset, which is acceptable for our application. It is of course possible to increase the number of PCs to retain more of the variability, but this comes at the expense of increased computational complexity. By limiting the number of PCs to six, we strike a balance between computational efficiency and the accuracy of the PC representation of the dataset, in terms of its ability to represent the covariance of the data.
We fit the PC model using a training dataset that is approximately unbiased in terms of spatial coverage. First, we divide the domain into $10°$ by $10°$ bins and select $f = N_{max} * \cos(\theta - \theta_0)$ profiles from each bin, where $N_{max} = 500$, $\theta$ is latitude, and $\theta_0 = 45°S$ is a reference latitude. The cosine factor help ensure that the reduced area at higher latitudes is taken into account. By using this approximately area-uniform training dataset, we offset the effect of the spatial bias in coverage on both the PC model and on the PCM (Fig. 2). As an alternative approach, we also tried a kernel-based PCA method, but the results were
nearly identical.

The six-component PC representation of the system features functions of both conservative temperature and absolute salinity (Fig. B1). Many of the temperature functions feature near-surface temperature inversions; they all feature a rapidly-changing near-surface transitioning into a more slowly-changing subsurface, as one might expect in ocean profile data in the top 1000m. Some of the salinity functions also feature such variation, although they are relatively better characterised by more gradual
changes between the near-surface and the subsurface. Using this PC representation, we reduce the dimensionality of the data to six PC coefficients. The distribution of those PC coefficients in six-dimensional space shows that a multi-dimensional Gaussian representation is an appropriate way to statistically model this dataset (Fig. B2). It is in this abstract, six-dimensional PC space that we use GMM to find patterns of coherent variability.

## B2 Selecting the number of classes

Since oceanographic profile data is highly correlated in space and time, it is unlikely that any classification approach would be able to cleanly and unambiguously identify groups or structures within the dataset that do not have at least some overlap. As such, we approach this classification task with the knowledge that any distinction between ocean profiles will retain some ambiguity. That being said, there are statistical tools that can be used to offer some guidance for making this decision. Two commonly-used criteria are the Bayesian Information Criterion (BIC) (Eq. B1) and Akaike Information Criterion (AIC)
(Eq. B2). Broadly speaking, they both contain a term that measures the agreement of the model to the data, and they both have a penalty term that discourages overfitting. Ideally, one would find a clear minimum in both BIC and AIC at the ideal value of





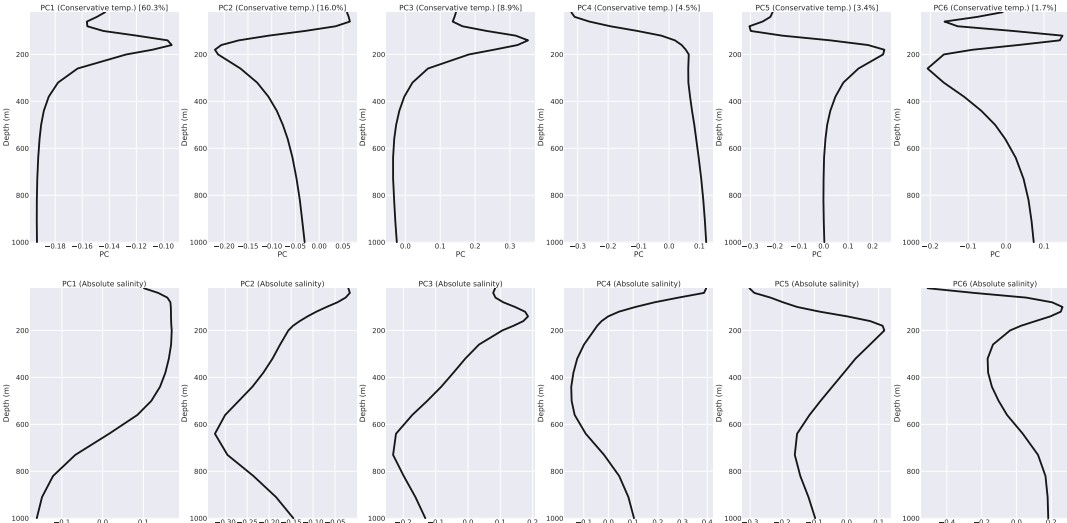

**Figure B1.** The six principal components (PC) in both temperature and salinity that are used for dimensionality reduction in this study. Each T-S profile is represented as a linear combination of these six principal components. The percent variance that is statistically explained by each principal component is shown in each panel title.

$K$. In practice, it is rare to find an unambiguous minimum, due to the highly correlated nature of the data (e.g. Sonnewald et al. (2019), Jones and Ito (2019)). BIC and AIC can be expressed as:

$$
\begin{aligned}
\mathrm{BIC}(K) &= -2\mathcal{L}(K) + \eta_f(K)\log(n), & \text{(B1)} \\
\text{with} \quad \eta_f(K) &= K - 1 + KD + \frac{KD(D-1)}{2}, \\
\mathrm{AIC}(K) &= 2K - 2\mathcal{L}, & \text{(B2)}
\end{aligned}
$$

where the log-likelihood is expressed as:

$$
\mathcal{L} = \log[\mathbb{P}(X)] = \sum_{n=0}^{N-1} \log\left(\sum_{k=1}^{K} \lambda_k \, \mathcal{N}(\boldsymbol{x}_n \, ; \, \lambda_k \, , \, \boldsymbol{\mu}_k \, , \, \Sigma_k)\right) \tag{B3}
$$

Above, $\mathcal{L}$ is a measure of likelihood, $\eta_f$ is the number of independent parameters to be estimated, and $N$ is the number of profiles used in the BIC/AIC training.

In addition to BIC and AIC, we also consider the silhouette coefficient, which is a measure of intra-cluster distance ($a$) and the mean nearest-cluster distance ($b$) for each profile (Rousseeuw, 1987). The silhouette coefficient is then $(b-a)/\max(a,b)$. The coefficient has values in the range [-1,1], where values near -1 indicate that a profile has been assigned "incorrectly" (i.e. it is more similar to a different profile), values near 0 indicate that there is overlap between the clusters, and values near 1 indicate relatively unambiguous classification within the context of the model being used. When working with ocean profile





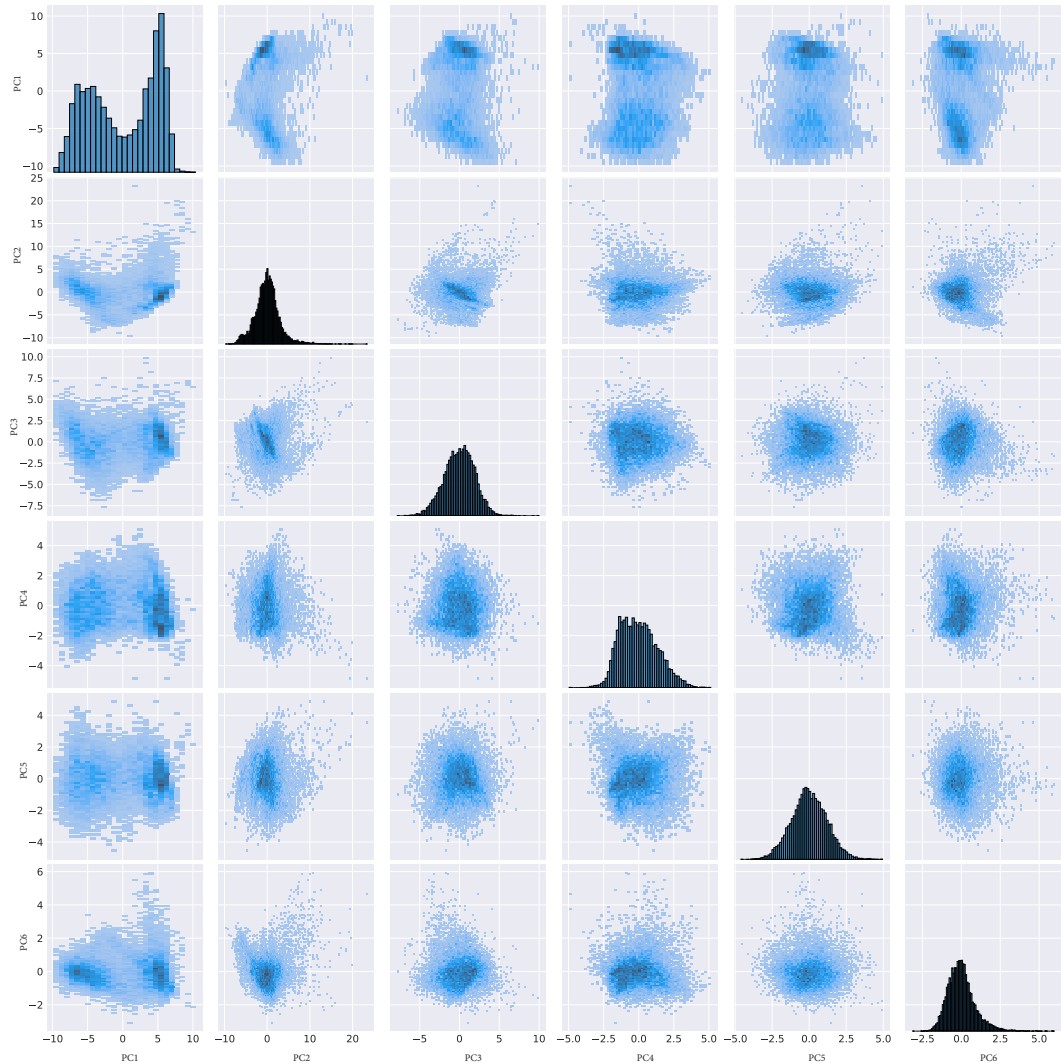

**Figure B2.** The distribution of profiles in PC space. Each row and column represents one of the PCs. The diagonal terms are distributions for each principal component. Diagrams such as this one can be useful for determining which statistical model should be used to describe the data.

data, which tends to be highly correlated, we do not necessarily expect values close to 1; the silhouette score that is considered "acceptable" varies with dataset and application.

We applied BIC, AIC, and the silhouette score to PCMs of the near-Antarctic data using values of $K$ between 2 and 19. For each value of $K$, we carried out the process 20 times. When applied to the PC representation of the near-Antarctic data, the combination of BIC, AIC, and silhouette score together indicate that the number of classes lies between $K = 3$ and $K = 8$ (Figure B3). We find a clear decrease in slope at $K = 3$ across all three metrics, which may be referred to as an "elbow". As

 

such, the simplest statistical representation of our model consists of three classes, which are physically representative of (1) the Weddell Gyre, (2) the circumpolar regime, and (3) a transition class between the two. By increasing the number of classes to $K = 4$, we see the distinction between the core of the Weddell Gyre and a class along its edge. This increase from $K = 3$ to $K = 4$ is favourable in terms of BIC and AIC, which slightly decrease (indicating greater likelihood), but at the expense of a decrease in silhouette score (indicating more overlap between the classes). Although further increasing $K$ to values greater than 4 improves AIC slightly, it does not improve BIC by much, and it comes at the expense of a decreased silhouette score (worsening overlap). Broadly speaking, the interpretability of our PCM worsens as the silhouette score decreases, as it becomes increasingly difficult to interpret the profile types as being somewhat distinct from one another. For values larger than $K = 8$, BIC begins to increase, indicating that we are in danger of overfitting the data in this regime. As a result of this analysis, we select $K = 4$ as the number of classes in our PCM, striking a balance between the complexity of our representation (i.e. the ability to distinguish the gyre core from the gyre edge) and its interpretability. For completeness, we include Jupyter notebooks for the $K = 3$ and $K = 8$ applications in the archived repository (Jones, 2022).

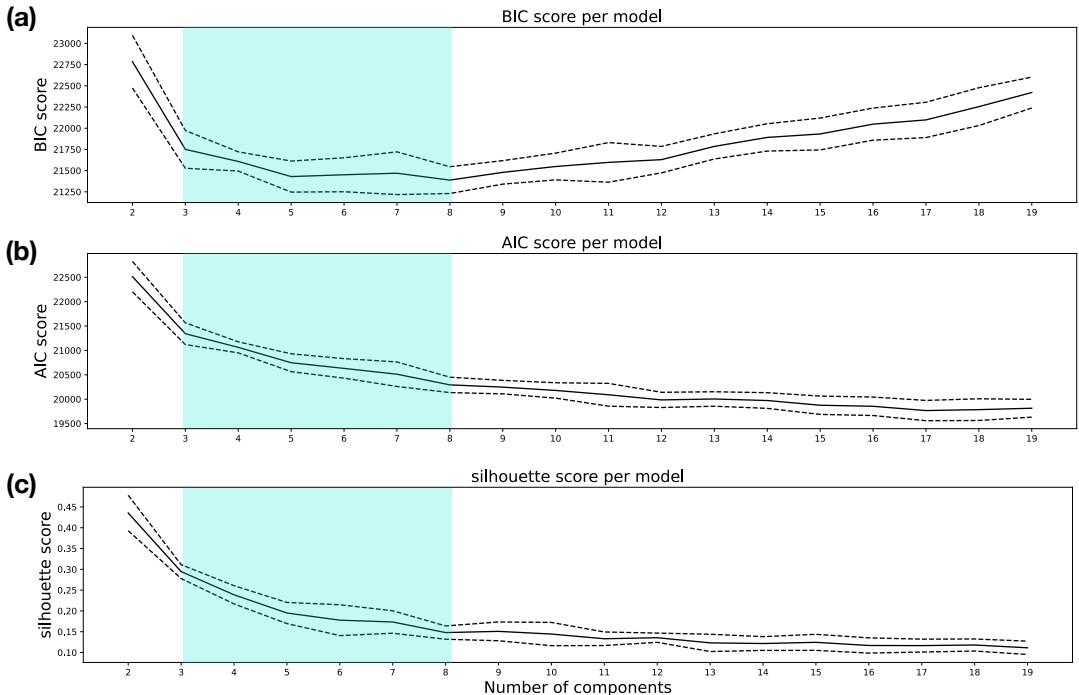

**Figure B3.** Statistical criteria used to guide the selection of the number of components in the GMM algorithm. Shown are the (a) Bayesian information criterion (BIC), (b) the Akaike information criterion (AIC), and (c) the silhouette score. For each $K$ value, these quantities are estimated using 20 random samples from the training dataset. The solid line represents the mean value, and the dashed lines indicate one standard deviation on either side. The light green shading indicates the range of classes that we suggest could be supported by these criteria when considered jointly.





## B3   Profile classification models

Unsupervised classification refers to a broad set of techniques that attempt to sort/group/label a dataset that has not already been sorted/grouped/labelled in some way. As a specific instance of unsupervised classification, Gaussian Mixture Modelling (GMM) attempts to statistically model the dataset under consideration using a set of multidimensional Gaussian functions (McLachlan and Basford, 1988). When GMM is used to sort ocean profiles, it is sometimes referred to as profile classification model (PCM), for example as seen in Maze et al. (2017). A PCM is "trained" or "fit" by iteratively adjusting the means and

covariances of the Gaussian functions, typically using an expectation maximisation approach. We offer a bit more mathematical detail below, adapted from the appendices of Thomas et al. (2021).

The GMM method attempts to represent the underlying data distribution using a set of $K$ Gaussian functions in $D$ dimensions (in our case $D = 3$):

$$\mathcal{N}\left(\boldsymbol{x}; \boldsymbol{\mu}_k, \Sigma_k\right) = \frac{\exp\left[-\frac{1}{2}\left(\boldsymbol{x} - \boldsymbol{\mu}_k\right)^T \left(\Sigma_k{}^{-1}\right)\left(\boldsymbol{x} - \boldsymbol{\mu}_k\right)\right]}{\sqrt{(2\pi)^D \|\Sigma_k\|}}, \tag{B4}$$

where $\boldsymbol{x} \in \mathbb{R}^{D \times 1}$ is a vector in the PC space, $\boldsymbol{\mu} \in \mathbb{R}^{D \times 1}$ is the center of the Gaussian distribution expressed in vector form, $\Sigma_k \in \mathbb{R}^{D \times D}$ is the covariance matrix, and $|\Sigma_k|$ is its determinant. The covariance matrix determines the orientation of the Gaussian ellipsoids in PC space. We statistically represent the dataset, using the following probability distribution:

$$\mathbb{P}(\boldsymbol{x}) \approx \sum_{k=1}^{K} \lambda_k \, \mathcal{N}\left(\boldsymbol{x} \, ; \, \boldsymbol{\mu}_k \, , \, \Sigma_k\right), \tag{B5}$$

where $\lambda_k$ is the weight associated with the $k$-th Gaussian. The GMM process iteratively adjusts $\lambda_k$, $\boldsymbol{\mu}_k$, and $\Sigma_k$ to decrease the

model-data misfit, which is called expectation maximisation, described in more detail in the next paragraph.

The first $K$ clusters are created randomly. Next, the set of Gaussians is iteratively adjusted (Eq.s B6, B7 and B8) until it reaches a local minimum in the cost function. The expectation of the model given the data is increased by updating the weights $\lambda_k$, means $\boldsymbol{\mu}_k$, and covariance matrices $\Sigma_k$ in the following way:

$$\lambda_k{}^{(t+1)} = \frac{1}{N} \sum_{n=1}^{N} \mathbb{P}\left(c_n = k \mid \boldsymbol{x}_n \, ; \, \{\lambda_k \, , \, \boldsymbol{\mu}_k \, , \, \Sigma_k\}^{(t)}\right), \tag{B6}$$

$$\boldsymbol{\mu}_k{}^{(t+1)} = \frac{\sum_{n=1}^{N} \mathbb{P}\left(c_n = k \mid \boldsymbol{x}_n \, ; \, \{\lambda_k \, , \, \boldsymbol{\mu}_k \, , \, \Sigma_k\}^{(t)}\right) \boldsymbol{x}_n}{\sum_{n=1}^{N} \mathbb{P}\left(c_n = k \mid \boldsymbol{x}_n \, ; \, \{\lambda_k \, , \, \boldsymbol{\mu}_k \, , \, \Sigma_k\}^{(t)}\right)}, \tag{B7}$$

$$\Sigma_k{}^{(t+1)} = \frac{\sum_{n=1}^{N} \mathbb{P}\left(c^n = k \mid \boldsymbol{x}^n; \{\lambda^k, \underline{\mu}^k, \Sigma^k\}^{(t)}\right) \cdot \left(\boldsymbol{x}_n - \boldsymbol{\mu}_k{}^{(t+1)}\right)\left(\boldsymbol{x}_n - \boldsymbol{\mu}_k{}^{(t+1)}\right)^T}{\sum_{n=1}^{N} \mathbb{P}\left(c_n = k \mid \boldsymbol{x}_n; \{\lambda_k, \boldsymbol{\mu}_k, \Sigma_k\}^{(t)}\right)}, \tag{B8}$$

where $c_n$ is the classification of the $n$-th cluster which could be any one of the $K$ clusters. The GMM algorithm repeats this process until the parameters have converged.





As discussed in section B1, we use a spatially unbiased training dataset to estimate the hyper-parameters of our GMM, ensuring that the resulting PCM is generally applicable across the entire spatial domain, rather than being specifically tuned to areas where there happen to be a large number of profiles (e.g. along ship track repeat sections).

Each profile is assigned a posterior probability distribution across the $K$ clusters (Eq. B9). This uncertainty information is one of the useful features of GMM. The probability takes the form:

$$\mathbb{P}\left(c_n = k \mid \boldsymbol{x}_n \, ; \, \lambda_k \, , \, \boldsymbol{\mu}_k \, , \, \Sigma_k\right) = \frac{\lambda_k \, \mathcal{N}\left(\boldsymbol{x}_n \, ; \, \boldsymbol{\mu}_k \, , \sum_k\right)}{\sum_{k=1}^{K} \lambda_k \, \mathcal{N}\left(\boldsymbol{x}_n \, ; \, \boldsymbol{\mu}_k \, , \Sigma_k\right)}. \tag{B9}$$

To label a dataset, each profile is assigned a label from the cluster that it would be the most likely to be generated by, in a statistical sense (Eq. B10):

$$\mathcal{C} = \arg\max_k \left(\mathbb{P}\left(c_n = k \mid \boldsymbol{x}_n \, ; \, \lambda_k \, , \, \boldsymbol{\mu}_k \, , \, \Sigma_k\right), \, 1 : k\right) \tag{B10}$$

## B4 PCM applied in PC space

The four-class PCM identifies (1) a circumpolar class, (2) a transition class between the circumpolar waters and the Weddell Gyre, (3) waters on the edge of the gyre, and (4) waters in the core of the gyre. These clusters are identified in a six-dimensional PC space, in which the dimensions are the principal component coefficients and each profile is represented by a single point. In a 3D projection of the six-dimensional PC space, which statistically explains about 85% of the variability, we see a covariance structure that may be described as "two wings and a bridge between them". The four-class model highlights these two "wings", namely a circumpolar wing and a gyre wing, separated by a transition "bridge" between the two wings (Fig. B4). If we imagine a stack of water masses transforming from more circumpolar-type to more gyre-type waters, this transformation would be represented by crossing the "bridge" from the circumpolar wing to the gyre wing. Much of the variability is contained in the first principal component (roughly 60%), which represents aspects of large-scale salt stratification.

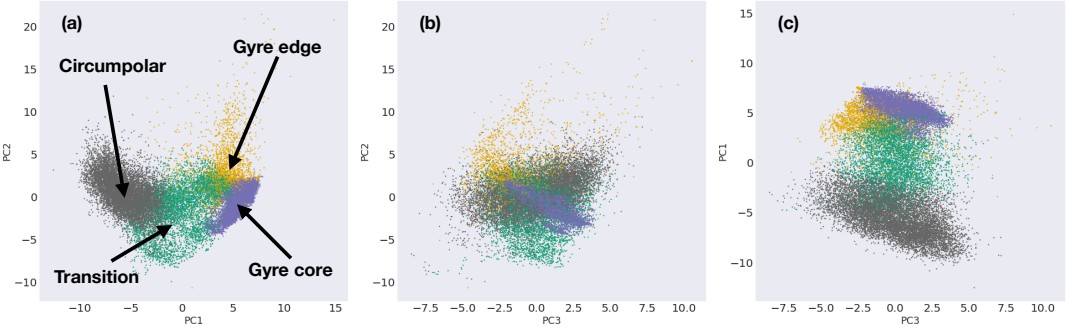

**Figure B4.** Distribution of profiles in 2D slices of the six-dimensional principal component (PC) space through combinations of the first three PCs. Each point in six-dimensional PC space represents a combined temperature and salinity profile. The color scale indicates the class into which each profile has been sorted by the GMM algorithm, including the circumpolar, transition, gyre edge, and gyre core classes.



*Author contributions.* DJ designed the study, wrote the software to carry out the analysis, and created the figures. DJ, MS, and IR collaborated on methods. SZ provided the cleaned, processed profile dataset and the stress-driven upwelling estimate. UH, IR, MM, AM, and AN-G

contributed to the interpretation of the classes and helped put the results into the wider context of the literature. All authors contributed to the writing of the manuscript.

*Competing interests.* The authors declare that they have no competing interests

*Acknowledgements.* DJ and UH are supported by a UKRI Future Leaders Fellowship (MR/T020822/1). SZ and AM are supported by the EU Horizon project SO-CHIC (No. 821001). We thank Erin Thomas and Earle Wilson for discussions that helped improve the quality of this

work. We used the scikit-learn Python module for unsupervised classification (Pedregosa et al., 2011).



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
