# Peer review of "Unsupervised classification identifies coherent thermohaline structures in the Weddell Gyre region"

_EGUsphere, 2022_

## Author Comment (AC1)

**Response to RC1**

Below, please find the reviewer's comments in blue and our replies in black.
* * *
This manuscript applies a method of unsupervised machine learning called a profile classification model (PCM) to ocean profile observations in the Weddell Gyre region with the goal to identify and classify areas, or sub-regions, within the Weddell Gyre that share similar temperature and salinity characteristics.

This manuscript clearly highlights how unsupervised classification schemes, such as PCM, can be powerful tools when applied to poorly sampled regions as they are able to identify patterns within highly complex data with no user input. Importantly, the manuscript stresses that PCM is *a complementary technique* in addition to other types of analysis techniques and confirms previously known thermohaline structures as well as sheds new light on more subtle thermohaline patterns within the Weddell Gyre. The authors use PCM to identify and analyze four categories of ocean profiles within the Weddell Gyre as follows: i) the circumpolar class, ii) a transition class, iii) a gyre edge class, and iv) a gyre core class.

This manuscript is clearly written and highlights a powerful, yet under-utilized technique in oceanography and climate science. I think this work is very interesting and I recommend it is accepted for publication after the following concerns are addressed and several modifications are made.

We are glad that you found the manuscript to be clearly written and interesting; thank you for your thoughtful, detailed, and constructive feedback. We have attempted to incorporate your detailed suggestions into our manuscript, and we hope that you find the revised work to be suitable for publication.

**Specific Comments**

-The authors stress one of the benefits of the PCM method is that it identifies "*both expected and underappreciated* structures" (line 499). However, the results are not clear about which structures are the novel, underappreciated, or previously unknown ones. The manuscript would benefit from a short discussion or clarification on which individual results from the PCM technique are the most critical or important to the research community and which results simply confirm already known patterns.

Yes, we agree that this was unclear in the original draft. On further reflection, we removed the somewhat subjective phrase "expected and underappreciated structures" from the manuscript, since expectations and appreciation may vary between researchers, depending on their particular expertise. However, we have attempted to better emphasize how our results compare with current understanding (see the revised results and discussion

sections). In particular, there is now a summary of key refinements to current understanding near the start of the discussion section.

-The description of the training process for the PC model would benefit from more details. The spatial bias is carefully considering in the training process, however the data contains significant temporal biases as well. Summer months are more heavily observed, as well as a general pattern of increasing observations through time (with spikes in recent years as well as around year 2010). How are the seasonal and annual temporal biases accounted for in the training process? What impact may this have on the results?

We agree that the original description of the training was lacking detail. We have expanded this section, which is now in Appendix B2. Also, yes, we agree that the seasonal and annual bias in training is a limitation of our current method. We discuss this in subsection 4.6 in the new draft. Overall, because we are trying to build a climatological picture of the Weddell Gyre region, we do not expect this bias to drastically alter our results. However, this is clearly an area where further investigation would be useful.

Additionally, it is not mentioned how large the training data set is, or what the 'training' process looks like for the PC model. Are the final PCM conclusions sensitive to how the PC model is trained?

We have expanded this section - see what is now Appendix A. Our sensitivity tests suggest that the PCM is robust with respect to how the training is done, considering both the principal component step and for the GMM step. We tried increasing the number of principal components to 10 and saw no appreciable difference in the results - this is likely because the existing six-component PC captures 95% of the variability. We also saw little difference between using the spatially biased dataset and the spatially unbiased dataset, except for some shifts around the Prime Meridian, which is co-located with two GO-SHIP repeat sections and is therefore overrepresented in the spatially biased dataset. Furthermore, the size of the training dataset did not substantially alter our results when varied between using 50% and 100% of the full profile dataset.

-Section 3.2-3.4 Figure 6/7/8 – Are these figures showing the mean of all individual profiles assigned each class at every spatial grid box? Why do you show the mean for these metrics, yet describe the profile classifications with the median? How sensitive are these metrics or the profile classifications to outliers?

Yes, this was due to a technical limitation with xarray plotting features. The xarray package supports calculating the mean at each spatial grid box, but calculating other quantities is more difficult at present. We do not expect our results to be affected by using both of these metrics, since we are only attempting to give indicative values for each class, as opposed to rigid values, i.e. we don't define the classes by these statistics.

Furthermore, did you look at the seasonal variations in space for the depth of mixed layer depth/minimum/ maximum temperature? You suggest the patterns represent deep winter convection in the shelf waters, yet the data is averaged over time for each grid box, and the observations contain more observations during summer months. Figure 10 is helpful to understand the seasonal variability of the classes – but it would be interesting to also analyze the spatial distribution of the depth metrics by season.

Thank you for this suggestion. We agree that the spatial variability of the seasonal cycle is important to consider. To address this, we have expanded the "4_seasonal_analysis" Jupyter notebook to include seasonal plots. We found that the seasonal cycle for each class displayed relatively little spatial variability, with a few key exceptions:

| Quantity | Class | Note |
|---|---|---|
| MLD | Circumpolar | Summertime values shallower in western part of domain |
| All | Transition | CDW intrusion most apparent in summer and autumn, partly due to seasonal bias in sampling |
| MLD | Gyre core | Winter and spring values especially deep in the south |
| Depth of PT max | Circumpolar | Summer and autumn values are shallower in western part of domain away from the Antarctic Peninsula, corresponding to lower latitudes |
| Depth of PT min | Circumpolar | North-south contrast in depth is especially apparent in winter and spring, i.e. shallower depths at higher latitudes as expected. Winter water is shallower at higher latitudes and deeper at lower latitudes |
| Depth of PT min | Transition and Gyre edge | Especially deep summertime and springtime values in the western part of the domain |

Overall, the spatial structure of the seasonal cycle mostly highlights the gradients already seen in the climatological average. We have added a paragraph and a new figure on this topic in section 3.5.

(cont'd) For example, are the MLD and min/max temperature depths distinct in wintertime vs summer over the near coast shelf?

There are seasonal contrasts in the transition and gyre edge classes in the near-coastal shelf areas just off the Antarctic Peninsula. We see this for MLD, the depth of the minimum temperature, and the depth of the maximum temperature.

Do we even have observations in those regions in the wintertime to identify known wintertime signatures with this method?

Although there is a seasonal sampling bias, with associated spatial sampling biases, the distribution of profiles covers the regions examined in this study (Figure R1). There are some gaps in wintertime open ocean sampling just east of the Prime Meridian, as well as a persistent gap just east of 40°E. More sampling in these regions, and an eventual revision of our clusters based on improved data, would of course be beneficial.

[Figure]

Figure R1. Profile counts in 1° latitude-longitude bins, grouped by season.

-Line 351-351/Figure 12: Part A: The strongest upwelling does appear to be co-located with the circumpolar and transition classes in most of the domain, however there is some strong upwelling in the far western part of the domain (between 40-60W, south of the SBDY) that do not seem to overlap with the circumpolar or transition class profiles, and seems to overlap more with gyre edge profiles, yet the upwelling here is stronger than the general large-scale gyre class upwelling. Do you have an explanation why this region seems to be unique?

Thank you for your thorough investigation of the upwelling region. We believe that the region that the reviewer is referring to is in the northwest Weddell Sea, off the Powell Basin/south to South Orkney (highlighted in Figure R2(a) with the green box). It is interesting that the upwelling signature extends south of SBDY into the gyre edge. And we believe that it is likely due to the distribution of sea ice, sea ice drift velocity, and the resultant surface stress exerted onto the sea surface.

[Figure]

Figure R2. (a) Ekman velocity. (b) ice-ocean stress curl overlaid with ice stress vector. (c). air-ocean stress curl overlaid with air-ocean stress vector. Note that the stress curl and stress vector are all incorporated with the sea ice distribution, hence the weak air-ocean stress over the region with sea ice coverage. (d) ice drift speed overlaid with ice drift vector. (e) ice drift speed gradient over laid with ice drift vector. Contours in all subpanels are climatological sea ice concentration distribution. Each field is climatological.

The Ekman upwelling velocity at the base of Ekman layer is,

$$w_{ek} = \frac{1}{\rho_o}\left[\nabla \times \left(\frac{\tau}{f}\right)\right].$$

Here, $\tau$ is the ocean surface stress, which is determined by both the sea ice drift and wind (Dotto et al. 2018):

$$\tau = \tau_{aw} * (1 - SIC) + \tau_{iw}SIC,$$

where $\tau_{aw}$ is the wind stress and $\tau_{iw}$ is the ice stress. We compared the ocean surface stress curl component associated with ice drift and wind (Figure R2b, c), and it is clearly shown that the ocean surface stress curl is dominated by the ice stress in the gyre edge and gyre core. The highlighted region is characterized by strong negative curl (cyclonic) that leads to a strong localized upwelling pattern. This region is on the edge of a region of year-round sea ice coverage and also a region where the sea ice drift starts to accelerate along with the gyre circulation. This ice drift speed acceleration signature can be seen in climatological ice drift speed map (Figure R2d) and also the ice speed gradient map (Figure R2e), which can contribute to the upwelling signal highlighted by the reviewer. Overall, the strong upwelling signal mostly falls in the transition class, but this particular pattern is of the same magnitude as the rest of gyre edge/core classes.

- Line 351-351/Figure 12:  Part B) If more observations existed in the near coast downwelling region, would you expect to identify an additional 'near coastal' class in this region? Would the exceptionally large seasonal cycle in vertical mixing in the near coast shelf region impact the results?

At present, the northern and southern branches of the gyre edge class do not separate into separate classes when *K* is increased. This suggests that there are enough structural similarities between the two regions, and perhaps even an advective/dynamical connection, to warrant them being placed in the same class. However, it is of course possible that including more measurements along the near-coastal downwelling region may identify this region as a separate class. One might consider targeting a shallower pressure/depth range and perhaps including seal-tagged profiles to differentiate this structure from the other classes.

- Section 4.6: Figure 14 shows several profiles which lie separately from both the transition class and gyre core class (in PC space) – this grouping is comprised primarily of both transition class and gyre core. Is this separation in PC space meaningful? Does this grouping have some traits in common that results in grouping them together in PC space? For example, are they co-located in space or time within the Weddell gyre region? Or do they have certain temperature/salinity traits that can be attributed to specific PC's in common so that they are clustered and isolated in this PC space, yet are categorized in different classes? Does increasing the number of classes used in the PCM change how these 'isolated' profiles are categorized?

We apologize for the confusion - this figure shows the data in t-SNE space, not the data in PC space. We have revised and expanded this section, including a revised figure that displays the t-SNE transformation using four different parameter values. We hope that this has improved the clarity of this section.

The grouping that you pointed out is still present in our revised t-SNE, but it is not separate from the wider distribution. So, the separation was neither robust nor meaningful. Upon further investigation, we found that this small grouping consists of profiles from the high-latitude, far-eastern portion of the domain, where the transition class and the southern extent of the gyre edge class overlap (in the GitHub repository associated with this paper, see the Jupyter notebook "8_examine_tSNE" for details). Increasing the number of classes does not change how the small grouping is classified (in the GitHub repository associated with this paper, see Jupyter notebook "2d_classify_antarctic-K8" for an example).

- Line 584: Please clarify. What process is applied 20 times for each value of K? It seems that the process of applying PCM can produce multiple realistic results (a different answer for any given iteration). Why were 20 iterations chosen? Are the results sensitive to the number of iterations?

Thank you for correctly pointing out that there was not nearly enough detail in this section. We have added extra details about this process. Specifically, in what is now Appendix A, we have added the following text explaining the process and why we used 20 iterations:

"For each value of $K$, we fit 20 different GMMs using randomly drawn 1,000-profile subsets of the training dataset. We used 20 different subsets because this empirically gave us stable statistics, i.e. doubling this to 40 made no appreciable difference on the distributions of BIC, AIC, and the silhouette score."

To answer your question specifically, no, the results do not appear to be sensitive to the number of iterations.

On PCM producing multiple realistic results: it is true that using different training subsets can produce different PCMs. By using different subsets of the spatially-unbiased training dataset as described in the text added above, we are examining the variability that is associated with the distribution of the training dataset. The resulting statistics help us to evaluate the generality of our GMM. As we argued in this section, $K=4$ is a generalisable and robust choice given this statistical distribution of BIC, AIC, and silhouette score values. Increasing $K$ up to $K=8$ may be defensible if greater detail is needed, but this will come at the expense of interpretability and generalisability.

Technical corrections

- The grey profiles on the figures (for example: Fig 3; Fig 10) are very difficult to see. Recommend a darker shade of grey.

Yes, thank you for this suggestion. This was due to our approach of plotting 18% of the profiles for each class, which of course meant that some plots had more lines than others, leading to uneven visibility across the subplots. We have revised our approach to include a random 1000 profiles (Fig. 3) or a random 400 profiles (Fig. 10) instead, producing more consistent visibility across the subpanels.

- Line 89: define ENSO

Done.

- line 135 – either define PF or spell it out.

Changed.

- Figure 2a: add units/label to color bar.

Done.

-Figure 2b: There are only 11 bars plotted in the monthly chart.

Thank you so much for pointing out this oversight - well spotted! We took the opportunity to review and adjust both distribution plots accordingly. The updated monthly plot now more clearly displays the bias towards observations in austral spring and summer. Additionally, the new distribution by year utilizes five-year bins, providing a more concise visualization. Thanks again for catching this - very helpful.

- Line 189: define WW

Done.

- line 245-255- Figure 5 is never referenced

Fixed. It is now referenced in section 3.2.

- line 251- the text specifies the 'mean' yet, the metric given is the median.

Fixed.

- line 290: typo – $^{o}$C?

Yes, good catch. Changed to °E.

The revised Fig. 11 now features:

- Larger x-axis and y-axis labels
- Larger x-axis and y-axis tick labels
- A single enlarged colorbar for the entire figure, instead of several smaller, identical colorbars
- Bigger colorbar tick labels and a bigger colorbar label
- Grey contours have been replaced with white contours for better contrast

Agreed, changed.

Fixed.

- The contents of Appendix A are very difficult to interpret since the methods are not presented until Appendix B. Add references to Eqns B1 and B2 for the AIC and BIC, and refer the reader to Appendix B.

Yes, this is a good point. We have switched the order of Appendix A and Appendix B, such that the methods are discussed first. We have also added the references to equations A1 and A2.

Fixed, thank you.

---

## Author Comment (AC2)

Response to RC2

Please find the reviewer's comments in blue and our replies in black.
* * *
This study applies an unsupervised classification technique, called profile classification model (PCM), to a profile data set in the Weddell Sea region. The authors highlight the importance of this technique as a powerful tool to identify spatially coherent structures in poorly observed regions, such as the Weddell Sea, in addition to other analysis techniques. Without any given spatial information, the PCM identified four spatially coherent classes: 1) circumpolar, 2) transition, 3) gyre edge and 4) gyre core, with sometimes overlapping but distinct properties of temperature and salinity for each class.

The authors emphasize that the PCM is able to reproduce expected structures from previous studies, but is also able to shine light on more subtle thermohaline structures in previously under-appreciated locations in the Weddell Gyre region.

The manuscript provides an interesting approach and a powerful technique to investigate thermohaline structures in poorly observed regions. The method is applicable to other branches of climate science and is thus a valuable contribution to the climate science community.

Thank you for your thoughtful and detailed response. We have attempted to address your comments below, and we hope that you find the revised manuscript to be improved and suitable for publication.

Before I recommend this study to be accepted for publication, I would like to see an updated version where specific concerns and questions have been addressed and modified.

Specific comments:
The manuscript would benefit from changes to the structure throughout. In some places the manuscript is quite repetitive, whereas in other places there is not enough discussion with previous literature. I think this is related to the large number of subsections (17 in total) in the manuscript that interrupt the flow and make it very difficult to follow at times. I suggest merging some of the subsections to increase readability and avoid repetition:

We apologise that you found the manuscript frustrating to read at times. With great respect to the reviewer, the use of repetition and subsections was very much intentional, and we would argue that both style elements are appropriate and suitable for how our intended audience is likely to engage with this paper. Our audience likely tends to read papers online, skipping around the paper and only skimming those sections that they find of interest. It is written with the casual reader in mind. The casual reader will hop around the paper, skim

sections, and look for take-home messages. Clearly-labelled subsections and repetition will help them navigate and understand the paper.

That being said, we took your feedback to mean that the paper's flow needed to be improved. As such, we did attempt to adapt the paper in specific ways following some of your suggestions. We sincerely appreciate the level of in-depth consideration that you gave this work; we took your suggestions seriously.

For example, we did eliminate one of the unnecessary subsection headers in the introduction in order to help it flow more smoothly. We decided to keep the methods-based subsection header (now 1.1) to clearly demarcate the circulation-based material from the methods-based material. We merged two of the discussion subsections that were both relevant to the WSC. We kept the rest of the subsections in the results and the discussion section for the reasons mentioned above. Overall, we also tried to make the subsection headings more informative and the transitions between subsections smoother. We hope that you find our revised paper to be an improvement in terms of readability.

Introduction: I suggest merging the subsection 1.1 and 1.2 in order to strengthen the storyline for the reader to follow. Simply, stating that this region is poorly observed is not strong enough.

We agree that the transition between these two subsections was too abrupt. Although we have kept the subsections for reasons mentioned above, we have attempted to make the transition to what is now subsection 1.1 a bit smoother.

Note that many scientists are not familiar with PCM, so I suggest introducing it first, before giving a literature review of which publications have used this method.

Yes, we agree that this section needed a bit more of an introduction to machine learning in general. The first four paragraphs of subsection 1.1 proceed from the more general to more specific. The first paragraph introduces unsupervised learning in general before citing a few recent oceanographic regime discovery applications. Here, the inputs/features are not necessarily profiles - they are terms of a steady-state budget equation or The second paragraph introduces the concept of profile classification, before citing a study that does this using self-organising maps. The third paragraph introduces PCM, and the fourth paragraph is a literature review of PCM applications in oceanography. We have modified this section to hopefully improve its clarity.

What specific questions are the authors going to address? What has not been investigated before? What is PCM and why using it? What is the aim of this study?

We apply the PCM technique to this newly-curated Weddell Gyre dataset to identify coherent regimes in temperature and salinity structure in this region, with the aim of refining our understanding of the area and generating new hypotheses for further investigation. This has not been done before because (1) the datasets have been too small until recently and (2) unsupervised classification techniques have not been used very much in oceanography until recent years. PCM is suitable for this application because it robustly and automatically groups profiles together into classes, allowing for the identification of coherent regimes in thermohaline structure, in a probabilistic fashion. We have added some text to the introduction that will hopefully clarify our aims and why we chose PCM.

Data and Methods: This section is quite repetitive. I suggest merging the subsections to avoid repetition. Can you elaborate more on how you use PCA to reduce the dimensionality? Or at least refer to the supplementary material if not discussed in the text?

Again with great respect, we believe that the subsections and repetition make it easier for the casual reader to navigate around the paper. Subsection 2.1 covers the dataset and how it was processed, and subsection 2.2 covers the PCM technique in more detail than had been given in the introduction. We would argue that this is a logical separation.

That being said, we completely agree that there was not enough detail about PCA. We have referred the reader to the supplementary materials (now Appendix A) for further details about how we used PCA.

There is an entire paragraph on the difficulty of classification as it becomes less interpretable the more complexity is added to the classes? Rather than explaining possible options can you specifically state how you overcame this issue?

In our view, this is a crucial paragraph. We wish to draw attention to this aspect of selecting the number of classes, especially for those who may wish to use this method or unsupervised classification in general. It is not only a technical issue to be overcome in our specific case: it is a general usage consideration for the entire field to be aware of, especially those who wish to use this technique with their data.

2: The label font sizes are inconsistent. a) The colorbar is not labelled. Does the colorbar indicate the number of profiles you have for each location? Why 2° latitude-longitude bins when you use 1° bins later on?

The font size inconsistency has been fixed in the new version of Figure 2. Yes, we totally agree that it makes more sense to use 1° bins throughout the paper – this has also been changed.

What are the advantages and disadvantages of the PCM overall?

PCM allows for the robust classification of profiles into profile types with very little prior input from the user. It also uses probabilistic information that can be further exploited to quantify boundaries between classes in a novel way. We have added some text to the introduction which hopefully clarifies this.

Results and Discussion: It is very difficult to jump between discussion and results due to the large number of subsections. I think the results and discussion would benefit from one another if they would be merged. Merging results and discussion avoids repetition, but also maintains readability and provides the ability for the reader to logically order the results and how this compares to previous studies.

We apologise that you found it difficult to navigate the manuscript. As discussed above, and emphasising again our great respect, our use of subsections and repetition is very much intentional. We have purposefully used the structure where one presents the key results in the results section and the limitations, wider context, and other aspects for consideration in the discussion section. Again, we again thank the reviewer for many specific, helpful suggestions, which we have attempted to incorporate into our revised manuscript.

A contour plot or similar would be really helpful to highlight where the classes overlap.

We experimented with different ways to plot the classes and where they overlap, but we did not find a plotting method that produced a visually clear result. We hope that showing the full spatial extent of the classes, together with the fronts, coastlines, and $f/H$ contours will make it possible to compare the spatial distributions of the classes.

L210-229: This paragraph contains a lot of speculation (lots of 'may') and would benefit from comparison to previous studies for justification. Are these results expected? Is it comparable to previous studies? What are the novel findings?

Yes, we agree that this comparison was a bit minimal in the previous version. We have revised this section to reference other studies and to highlight which of the findings might be considered novel, or at least which findings might be considered useful refinements of our understanding.

L244-246: This sentence is very unclear to me. Are you indicating that a large I-metric suggests regions of increased mixing or a transformation in profile types or both? Please clarify.

Apologies for the confusion. We have changed this to "mixing and transformation between profile types", as we did mean both.

L247-250: Can you specify what a low I-metric indicates?

Yes, we have added a bit more detail here as to what a low I-metric indicates in this context.

Furthermore, the authors emphasise that PCM highlights both expected and underappreciated structures and is thus a helpful tool to develop new research questions and hypotheses. It is unclear which structures are expected and which are novel or previously underappreciated. Which new findings did this method provide and which additional analysis steps are suggested to investigate the novel findings in more detail?

Yes, R1 had a similar comment. We agree that this was unclear in the original draft. On further reflection, we removed the somewhat subjective phrase "expected and underappreciated structures" from the manuscript, since expectations and appreciation may vary between researchers, depending on their particular expertise. However, we have attempted to better emphasise how our results compare with current understanding (see the revised results and discussion sections). In particular, there is now a summary of key refinements to current understanding near the start of the discussion section.

The authors further discussed seasonal variations within each class specifically with respect to vertical profiles and Theta-S diagrams. Did you also consider seasonal variations in mixed layer depth, signatures of WW (temperature minimum), signatures of CDW (temperature maximum) and how those vary spatially within the classes?

Thank you for this suggestion. We had not examined the spatial variation in the class properties by season, but we agree that this is an important consideration. As mentioned in our response to RC1, we have expanded the "4_seasonal_analysis" Jupyter notebook to include seasonal plots. We found that the seasonal cycle for each class displayed relatively little spatial variability, with a few key exceptions:

| Quantity | Class | Note |
| --- | --- | --- |
| MLD | Circumpolar | Summertime values shallower in western part of domain |
| All | Transition | CDW intrusion most apparent in summer and autumn, partly due to seasonal bias in sampling |
| MLD | Gyre core | Winter and spring values especially deep in the south |
| Depth of PT max | Circumpolar | Summer and autumn values are shallower in western part of domain away from the Antarctic Peninsula, corresponding to lower latitudes |
| Depth of PT min | Circumpolar | North-south contrast in depth is especially apparent in winter and spring, i.e. shallower depths at higher latitudes as expected. Winter water is shallower at higher latitudes and deeper at lower latitudes |
| Depth of PT min | Transition and Gyre edge | Especially deep summertime and springtime values in the western part of the domain |

Overall, the spatial structure of the seasonal cycle mostly highlights the gradients already seen in the climatological average. We have added a paragraph and a new figure on this topic in section 3.5.

How did you deal with the varying number of profiles per season (less profiles in winter months)? How robust are the results?

We agree that the seasonal bias in sampling is a limitation of our current method. We discuss this in subsection 4.6 in the paper. Overall, because we are trying to build a climatological picture of the Weddell Gyre region, we do not expect this bias to drastically alter our results. However, this is clearly an area where further investigation would be useful.

Minor comments:
Suggest removing preambles in all sections as they are mostly repetition to the manuscript structure you have already introduced in the introduction.

The preambles help orient readers who may be jumping around and skimming the paper. As mentioned above, we are writing with the casual reader in mind.

Adjust spelling as required by ocean sciences (British or American spelling not both)

We have attempted to use British spelling throughout. That being said, the lead author is hopelessly mixed in a blend of the two spellings. Although we have attempted to be consistent in this new version, we trust that the copyediting process will catch any remaining spelling oversights.

L3, L4, L23: Can you define extremely? Suggest giving values here.

We have removed the "extremely" from L3. In terms of L4, we think that "extremely remote" is clear from context. For L23, the phrase "below the surface freezing point" provides context.

Suggest adding more information on how this method is valuable to the wider community in the abstract.

We have made the last sentence of the abstract a bit more general and have added a bit more detail. Specifically, we have revised this to state that "PCM offers a useful complement to existing expertise-driven approaches for characterising the physical configuration and variability of oceanographic regions, helping to identify coherent thermohaline structures and the boundaries between them."

L84-96: This is just a list of references who have used PCM before. How is this relevant to this study? Note that at this point you have not introduced the method yet.

The paragraph preceding the original L84-96 introduces PCM in a conceptual way. We feel that the introduction is an appropriate place to discuss other studies that have deployed PCM, specifically because each one has pushed the frontiers of what is possible with this method. Along with each study, we mention the specific innovation delivered by that paper, which is relevant to the developing maturity of PCM as a technique. In addition, we hope that this paper will be read by those who may be interested in trying PCM. For those readers, a survey of recent applications is useful.

L96-99: Is additional motivation that PCM can be used to redefine boundaries and fronts? You don't mention it again? It would be really interesting if you could elaborate more on that.

Yes, we discuss this again in subsection 3.2. We have revised the subsection title to better signpost this work.

L189: Define Winter Water

Done

L208-209: How much does it deepen the further you go away from the gyre core?

Reporting the 25th, 50th, and 75th percentiles, the winter water depths range from (20m, 60m, 80m) in the gyre core, (20m, 80m, 100m) in the gyre edge, (60m, 80m, 120m) in the transition class, and (100m, 120m, 140m) in the circumpolar class. We have added these values to the text.

L221-223: Can you elaborate more on why these are indicators of e.g. Weddell-Scotia Confluence waters? Is this new information? Is this comparable to previous studies?

Yes, we believe that the revised subsection 4.3 makes this a bit clearer.

L247-262: Can you refer to the Figs. And subplots here?

Yes, thanks for catching this oversight. Added.

Equations B1 and B2. Define K.

Done, thanks for catching this omission. These equations, along with the definition for K, are now in Appendix A.

Figure comments
Overall, I suggest using different colormaps for Figures, showing different variables.
Consider changing the colormaps to increase contrast between contours and increase
visibility.

We make heavy use of the "cmocean" colormaps for oceanography because they are
perceptually uniform (Thyng et al., 2016). We also use the "cividis" colormap for sequential
data because it is also perceptually uniform, in addition to being colorblind friendly. Other
options that we tried were considerably worse or were inappropriate for sequential data. We
note that our colormap choices were tested electronically.

Fig 1: Can you describe the choice of arrow colors on this map? Did you choose the colors
based on temperature? If so, why do CDW and WDW have the same color? On a printed
version you can barely see the arrows underneath the geographic locations added to the
Fig. I suggest moving the all geographic locations labelled in the Fig. outside to achieve a
clear view. Further the font size of the x- and y-axis labels needs to be increased.

The original color scheme for the arrows was not based on temperature and was arbitrarily
chosen solely for visual separation. However, we have made revisions by using a colour
scheme recommended by colorbrewer2.org. We chose a 4-class diverging colour scheme
that is colorblind safe, print-friendly, and photocopy safe. We kept the original ASF colour,
but the new scheme should provide better visibility and clarity.

We apologise for the text boxes' opacity not producing the desired effect in the printed
version of the paper. We intended for the semi-transparent boxes to allow the text to stand
out without blocking the arrows completely. We have addressed this issue by repositioning
the text boxes to improve clarity.

Additionally, we have increased the font size of the axis labels to improve readability.

3: In section 3.1 you discuss the differences between each identified class. You specifically
mention the median depth of the temperature minimum. I suggest adding the median depth
as a horizontal line to each class to visually highlight the differences in each class.

The median depth of the temperature minimum does not vary enough between the classes
to be visually distinct on the scale of Fig. 3. However, on Fig. 3 we have added dotted lines
to indicate, for each class, the median depth of the temperature maximum.

In addition, as described in our response to RC1, we have changed the way in which we
select and plot the individual profiles for Figs. 3 and 10, leading to more consistent visibility
across the subplots.

On Figs. 4, 5, 6, 7, 8 the grey lines are hardly visible. Suggest using a darker shade of grey
and thickening the lines. I also suggest using a different colormap for each variable to
visually differentiate between the plots. The blue is very dominating, so other structures are
very hard to see.

We apologise that you found the grey lines to be hardly visible – is the author referring to the print version? They appear to be suitable on the electronic version, as far as we can tell. We intended the grey lines to be a visually "weak" element of the figures, providing a little dynamical context without overwhelming the other visual elements. We experimented increasing the line thickness and darkening the colours, but doing so causes the figures to become visually messy and hard to read. Should the article be accepted in OS, we will seek advice from the copyediting team on which line thickness and colour will achieve good visual results in both the print and electronic versions.

As mentioned above, we make heavy use of the "cividis" colormap because it is colorblind friendly, perceptually uniform, and sequential.

Figs. 7 and 8. Figures are much too small. Labels are too small.

Configuring these two plots is certainly a design challenge. We wish to have the depths and values plotted side-by-side for comparison; this works okay for the electronic version, because readers can zoom in as needed. But we agree that this makes them too small in the print version. If this article is accepted, we trust that the typesetting service will help us resolve this design challenge, perhaps by allowing these two figures to be rotated where they take up an entire page.

Figs. 9. You are using conservative temperature. Shouldn't it be Θ-S diagrams? Suggest changing this throughout.

Yes, thank you for catching this. Changed throughout.

Fig. 11 The background is too dark. I cannot read the isopycnal labels (much to small, contrast too low)

The background is the lowest value of the "cividis" colorbar that we wish to use due to it being colorblind friendly and perceptually uniform. However, we hope that you find the revised Fig. 11 to be improved. The revised figure now features:

- Larger x-axis and y-axis labels
- Larger x-axis and y-axis tick labels
- A single enlarged colorbar for the entire figure, instead of several smaller, identical colorbars
- Bigger colorbar tick labels and a bigger colorbar label
- Grey contours have been replaced with white contours for better contrast

Fig. 12. Suggest labeling the colorbars. In b) suggest adjusting the colormap. It is really hard to see the sea ice freezing line.

Yes, good suggestions, thank you. Done.

**Reference**

Thyng, K. M., Greene, C. A., Hetland, R. D., Zimmerle, H. M., & DiMarco, S. F. (2016). True colors of oceanography. *Oceanography*, 29(3), 10

---

## Author Response (AR2)

**Response to editor's decision**

Please find editor comments in blue and our responses in black.

Dear Dr. Jones and co-authors,
Thank you for submitting your revised version to Ocean Science. I would like to take the opportunity to thank both reviewers for their very thorough and constructive comments, as well as to thank the authors for their detailed and respectful responses. I find that most concerns were addressed by the authors and have some more minor issues that I noticed during my evaluation of the revised article.
The paper presents a comprehensive machine learning-based classification of oceanographic "provinces" from profiles from the Weddell Gyre region. Using unsupervised machine-learning may present a new analysis technique for many members of the WSDML community, especially in the context of an interdisciplinary special issue, which is why the level of detail used in this paper is appropriate.

We are glad that you found the paper to be comprehensive and within the scope for the special issue.

Nevertheless, the paper is overall quite long and as criticized by reviewer #2, the paper is divided into many subsections, which may create redundancy and decrease the text flow in certain sections. However, the authors state that the structure was chosen specifically to allow for easier browsing for the casual reader. While I can relate to the reviewer's concerns, I am generally happy to support the author's approach, as the structuring style is a matter of taste and preference.

Below are some minor comments and suggestions for figure improvements, which I encourage the authors to consider before acceptance of the paper.

Thank you for your helpful suggestions. Please see our response below.

Minor comments on text

Line 39: do you mean to say "30°E, 70°E"?

We have changed this for simplicity.

Line 40: 60°S

Yes, thanks for catching this.

Line 112: suggest to include the reference for the data collection here

Done.

Line 119: paper
Fixed.

Lines 124-125 and 198-204: In my opinion, these are unnecessary introductions to subsections, as was also pointed out by one reviewer. Please re-consider whether preambles are really needed.

Removed.

Line 78 and 224: „in latitude and longitude" seems unnecessary

Removed.

**Line 247: perhaps better say "thickness of the water column"**

Apologies, but I wasn't sure which phrase you were referring to here, so I didn't change anything.

Line 331: parenthesis and period missing

Fixed.

Line 372: „becomes" is used twice in this sentence

Yes, thanks. We removed one of them.

Line 446: an

Changed.

Section 3.5 in particular as well as other text passages: the word "relatively" is used somewhat inflationary

This is a great point – the word "relatively" really was overused in the original draft. I have removed or changed most of the instances of this word. Now it only shows up four times, which is relatively few (ha).

Suggestions for figure improvements:

Discrete colorbars: This may also be a matter of taste and hence suggestions rather than requests, but discrete increments generally provide more quantitative detail in the figures. While some of your figures employ discrete colorbars, this could be used to improve figs 2, 4-8 and 12 as well.

Thank you for this excellent and specific suggestion. We have replaced the colormaps with discrete ones.

Fig 3: Why are different colors used for the profiles?

Each class gets its own color. We use the same scheme for Figure 10 and Figure 15.

Fig11: the tickmarks on the colorbar do not quite agree with the increments

Updated. They now line up.

Fig 12,13: "latitude", "longitude" labels are missing

For consistency, because none of the other map plots have "latitude" or "longitude" labels, we left them off here as well.

Fig 13: remove right-hand y-axis labels

Done.

Fig 14: The panels are missing grids and need more detailed captions

Yes, good point. The caption was very minimal. This has been updated.

FigA1: very small fonts, consider using only one ylabel "depth (m)" on the left hand side and remove all other labels, as the panels overall share identical y-axes. Perhaps include titles with larger fonts inside of the panels

Updated.